# RODRIGUES NETWORK
# FOR LEARNING ROBOT ACTIONS

**Jialiang Zhang**[* ‡]
MIT, Stanford University

**Haoran Geng**[*]
UC Berkeley

**Yang You**[*]
Stanford University

**Congyue Deng**
MIT, Stanford University

**Pieter Abbeel**
UC Berkeley

**Jitendra Malik**
UC Berkeley

**Leonidas Guibas**
Stanford University

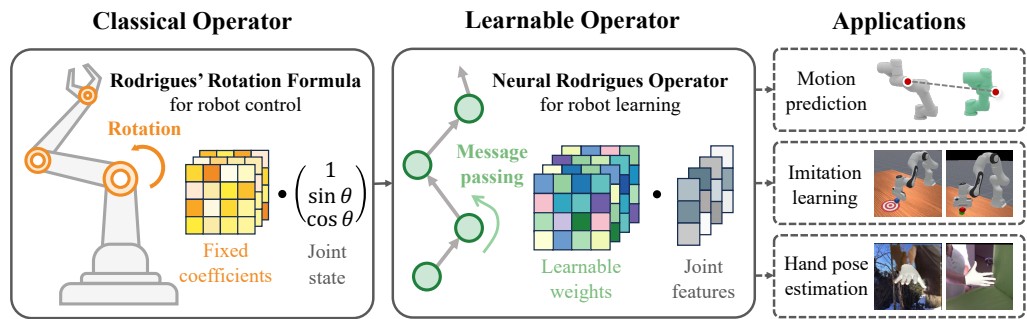

Figure 1: We introduce the **Neural Rodrigues Operator**, a learnable extension of the classical Rodrigues' Rotation Formula from robot control, where the original coefficients are replaced with trainable weights and joint angles are generalized to abstract features. Built upon this operator, the **Rodrigues Network** leverages the kinematic structure of articulated systems to advance a wide range of action-learning tasks.

## ABSTRACT

Understanding and predicting articulated actions is important in robot learning. However, common architectures such as MLPs and Transformers lack inductive biases that reflect the underlying kinematic structure of articulated systems. To this end, we propose the **Neural Rodrigues Operator**, a learnable generalization of the classical forward kinematics operation, designed to inject kinematics-aware inductive bias into neural computation. Building on this operator, we design the **Rodrigues Network (RodriNet)**, a novel neural architecture specialized for processing actions. We evaluate the expressivity of our network on two synthetic tasks on kinematic and motion prediction, showing significant improvements compared to standard backbones. We further demonstrate its effectiveness in two realistic applications: (i) imitation learning on robotic benchmarks with the Diffusion Policy, and (ii) single-image 3D hand reconstruction. Our results suggest that integrating structured kinematic priors into the network architecture improves action learning in various domains.

## 1 INTRODUCTION

We study the problem of understanding and predicting the actions of articulated actors. Articulated actors (*e.g.* robots (Shaw et al., 2023) and animated characters (Romero et al., 2022; Loper et al., 2023)) are systems that use multiple rotational joints to connect moving links. Their actions, including poses, motions, or control commands, are usually represented as values associated with all joints.

---

[*]Equal contribution
[‡]Work performed as a visiting student at Stanford University.

Learning with articulated actors usually involves predicting their actions while processing diverse sensory inputs. Such a problem lies in a wide spectrum of intelligent systems, from whole-body controllers (Moro & Sentis, 2019; Kuang et al., 2025; Geng et al., 2025), to dexterous grasp detectors (Duan et al., 2021; Xu et al., 2023; Wan et al., 2023; Zhang et al., 2024), to motion retargeting networks (Aberman et al., 2020).

These action data are inherently structured by articulated kinematics. This motivates us to design a neural network that leverages this structure as an inductive bias for better understanding and inference of actions. However, general architectures, such as MLPs and Transformers (Vaswani et al., 2017), treat actions as unstructured tokens, ignoring the kinematic relationships between joints. Previous works have exploited the connectivity of robot links through graph convolutions (Yan et al., 2018; Ci et al., 2019; Aberman et al., 2020) or masked attention (Sferrazza et al., 2024). Yet, architectures that directly exploit the computational patterns of articulated kinematics remain underexplored. This gap raises our central question: *Can we design a neural network for action learning that embeds articulated kinematics as an inductive bias?*

To answer this question, we draw an analogy to convolutional neural networks for images. Low-level 2D image features are spatially local and translation equivariant. Traditional pattern recognition exploits this structure using hand-crafted filters to detect edges (Canny, 1986) and corners (Harris et al., 1988). CNNs (LeCun et al., 2002) build on this idea by making filters learnable, adding nonlinearities, and using high-dimensional channels. This creates a deep learning framework that learns high-level semantic features while preserving the structural properties of classical image filters.

In a similar vein, we identify Rodrigues' rotation formula as the fundamental operator in articulated forward kinematics and transform it into a learnable form, which we call the **Neural Rodrigues Operator**. Specifically, we separate the entries in the Rodrigues' rotation formula into state-dependent parameters conditioned on the joint angles, and state-independent coefficients that only rely on the articulated structure. We then convert it into a neural operator by treating the state-dependent parameters as input features, and relaxing the state-independent coefficients into optimizable weights. We further extend it into a multi-channel operator that applies to higher-dimensional features rather than just 1D joint angles.

With our neural operator as the key component, we construct a complete network architecture, named the **Rodrigues Network (RodriNet)**, for encoding, understanding, and predicting articulated actions. It comprises three key components: a Rodrigues Layer for passing information from joints to links, built up our neural operator; a Joint Layer for passing information from links to joints; and finally, a Self-Attention Layer for global information exchange. We also introduce a global token for processing other task variables, such as perception inputs. Although starting with an operator in classical robotic theory, we end up with a deep neural network with modern designs while maintaining the structural bias in articulated kinematics, as illustrated in Figure 1.

We evaluate our approach through three sets of experiments: First, we demonstrate the strong expressivity of our network in representing forward kinematics and motions through toy experiments. Second, we showcase our effectiveness in realistic robot-learning scenarios with imitation learning on 5 robot manipulation tasks. Additionally, we show our network achieving state-of-the-art results in human hand pose estimation from images, where the articulated actor is no longer a robot, highlighting its applications in computer vision and graphics.

## 2 RELATED WORK: ARTICULATION-AWARE ROBOT LEARNING

Articulated robots can be naturally modeled as graphs, making graph convolution (Bruna et al., 2013; Niepert et al., 2016) a popular approach for processing articulated data. This has been widely applied in character animation for tasks like skeleton-based action recognition (Yan et al., 2018; Cheng et al., 2020; Song et al., 2022; Chen et al., 2021b), pose estimation (Ci et al., 2019; Choi et al., 2020; Zeng et al., 2021), and motion retargeting (Aberman et al., 2020). Graph convolution effectively captures link connectivity and spatial locality, but it does not explicitly incorporate articulated kinematics, which is an essential aspect of articulated action understanding. In contrast, our work introduces a novel operator derived from forward kinematics, providing networks with kinematics-informed inductive bias.

Transformer-based architectures (Vaswani et al., 2017) are widely used in policy learning (Brohan et al., 2022; Zhao et al., 2023; Shridhar et al., 2023), and recent work has explored incorporating structural bias via graph-aware positional embeddings (Hong et al., 2021) or masked attention (Sferrazza et al., 2024). However, these modifications do not fundamentally adapt the self-attention mechanism to suit kinematics. We instead use standard self-attention layers for network capacity, but rely on our kinematics-inspired operator for inductive bias.

Forward kinematics provides a deterministic mapping from joint space to Cartesian space, but integrating it into neural networks remains non-trivial. Prior methods have inserted analytical forward kinematics as a differentiable layer in neural networks (Villegas et al., 2018) to help them reason about the Cartesian results of the robot action. While this introduces kinematic awareness, it constrains the model's flexibility. Other methods apply Cartesian-space loss functions after computing forward kinematics on network outputs (Pavllo et al., 2020; Jiang et al., 2021; Liu et al., 2020), but these approaches do not alter the network architecture and are thus orthogonal to our focus. In contrast, our method derive a learnable operator from forward kinematics, thereby making the network kinematics-aware while maintaining the flexibility to learn high-level features.

# 3 NEURAL RODRIGUES OPERATOR

In this section, we derive the **Neural Rodrigues Operator** by making the Rodrigues' rotation formula learnable and more generalized.

## 3.1 BACKGROUND

**Articulated robots** In this paper's context, an articulated robot has a loop-free kinematic tree structure, with $D + 1$ rigid links $L_0, \cdots, L_D$ connected by $D$ one-DoF revolute joints $J_1, \cdots, J_D$. This definition encompasses a wide range of platforms such as robotic arms, dexterous hands, quadrupeds, and humanoids. The kinematic tree has a root, denoted as the base link $L_0$, which can be either fixed or free-floating. All links and joints have their local 3D frames. Each joint $J_j$ connects a parent link $L_{\mathrm{p}_j}$ with a child link $L_{\mathrm{c}_j}$, where a rotation axis $\hat{\boldsymbol{\omega}}_j \in \mathbb{R}^3$ in the joint frame defines the rotational motion of the child link relative to the parent link, as well as a fixed transformation from the parent link's frame to the joint's frame $\mathbf{T}_j \in \mathrm{SE}(3)$.

Given the kinematic structure, the configuration of the robot is then determined by the joint angles $\boldsymbol{\theta} = [\theta_1, \cdots, \theta_D] \in \mathbb{R}^D$, as well as the root pose $\mathbf{P}_0 \in \mathrm{SE}(3)$ if its base link is free-floating (not required for fixed-base robots). In most classical control systems, control commands are sent to the joints, specifying their joint angles, velocities, or torques, depending on the control modes.

**Forward kinematics** To obtain the position and orientation of all links, including the end-effector, given a set of joint angles, we apply forward kinematics. Below, we briefly outline its computation using homogeneous coordinates. We represent the pose of link $L_i$ as a homogeneous matrix $\mathbf{P}_i$ describing the transformation from the world frame to the link's frame:

$$\mathbf{P}_i = \begin{bmatrix} \mathbf{R}_i & \mathbf{t}_i \\ \mathbf{0}_{1 \times 3} & 1 \end{bmatrix} \in \mathbb{R}^{4 \times 4} \tag{1}$$

where $\mathbf{t}_i \in \mathbb{R}^{3 \times 1}$ is the position and $\mathbf{R}_i \in \mathbb{R}^{3 \times 3}$ is the orientation. Given the pose of the base link $\mathbf{P}_0$, the poses of the descendant links can be computed recursively from parents to children. More concretely, at joint $J_j$, using $\mathbf{T}_j, \hat{\boldsymbol{\omega}}_j, \theta_j$, we can compute the child-link pose $\mathbf{P}_{\mathrm{c}_j}$ from the parent-link pose $\mathbf{P}_{\mathrm{p}_j}$ through two transformations: (i) a fixed coordinate change $\mathbf{T}_j$ from the parent frame to the joint frame; (ii) a dynamic rotation $\mathbf{R}(\hat{\boldsymbol{\omega}}_j, \theta_j) \in \mathbb{R}^{3 \times 3}$ around axis $\hat{\boldsymbol{\omega}}_j$ of angle $\theta_j$ in the joint frame. Putting these together, the parent-to-children pose transformation is:

$$\begin{bmatrix} \mathbf{R}_{\mathrm{c}_j} & \mathbf{t}_{\mathrm{c}_j} \\ \mathbf{0}_{1 \times 3} & 1 \end{bmatrix} = \begin{bmatrix} \mathbf{R}_{\mathrm{p}_j} & \mathbf{t}_{\mathrm{p}_j} \\ \mathbf{0}_{1 \times 3} & 1 \end{bmatrix} \mathbf{T}_j \begin{bmatrix} \mathbf{R}(\hat{\boldsymbol{\omega}}_j, \theta_j) & \mathbf{0}_{3 \times 1} \\ \mathbf{0}_{1 \times 3} & 1 \end{bmatrix}. \tag{2}$$

Here, transformation (i) only depends on the robot's articulated structure and thus is fixed, and transformation (ii) is state-dependent with variable $\theta_j$. Therefore, we can abbreviate Equation 2 as $\mathbf{P}_{\mathrm{c}_j} = \mathbf{P}_{\mathrm{p}_j}(\mathbf{T}_j \tilde{\mathbf{R}}(\hat{\boldsymbol{\omega}}_j, \theta_j))$, where $\tilde{\mathbf{R}}(\hat{\boldsymbol{\omega}}_j, \theta_j) \in \mathbb{R}^{4 \times 4}$ is the homogeneous matrix of the rotation. Essentially, forward kinematics is a hierarchical composition of fixed coordinate transformations and dynamic rotations in the axis-angle representation with variables $\theta_1, \cdots, \theta_D$.

**Rodrigues' rotation formula** The Rodrigues' rotation formula tells how to compute the rotation matrix $\mathbf{R}(\hat{\boldsymbol{\omega}}, \theta) \in \mathbb{R}^{3 \times 3}$ from the axis-angle representation:

$$\mathbf{R}(\hat{\boldsymbol{\omega}}, \theta) = \mathbf{I}_3 + \sin\theta[\hat{\boldsymbol{\omega}}] + (1 - \cos\theta)[\hat{\boldsymbol{\omega}}]^2 \tag{3}$$

where $[\hat{\boldsymbol{\omega}}] \in \mathbb{R}^{3 \times 3}$ is the skew-symmetric cross-product matrix of the rotation axis $\hat{\boldsymbol{\omega}} = (\hat{\omega}_x, \hat{\omega}_y, \hat{\omega}_z)$.

### 3.2 Neural Rodrigues Operator with learnable parameters

Observe from Equation 3 that every entry in $\mathbf{R}(\hat{\boldsymbol{\omega}}, \theta)$ is essentially a linear combination of 1, $\cos\theta$, and $\sin\theta$, with fixed coefficients determined by the rotation axis $\hat{\boldsymbol{\omega}}$. Therefore, every entry in $\mathbf{P}(\hat{\boldsymbol{\omega}}_j, \theta_j)$ and thereby $\mathbf{T}_j \mathbf{P}(\hat{\boldsymbol{\omega}}_j, \theta_j)$ are linear combinations of 1, $\cos\theta_j$, and $\sin\theta_j$, with constant coefficients defined by the state-independent parameters $\mathbf{T}_j, \hat{\boldsymbol{\omega}}_j$ of joint $J_j$. Thus, we can re-parameterize Equation 2 as:

$$\mathbf{P}_{c_j} = \mathbf{P}_{p_j}(\mathbf{A}_j + \mathbf{B}_j \cos\theta_j + \mathbf{C}_j \sin\theta_j) \tag{4}$$

where $\mathbf{A}_j, \mathbf{B}_j, \mathbf{C}_j \in \mathbb{R}^{4 \times 4}$ are coefficient matrices that only depend on the robot's articulated structure. Based on this, we construct our **Neural Rodrigues Operator** for one single joint by replacing these fixed coefficients with learnable weights $W^{\text{bias}}, W^{\cos}, W^{\sin} \in \mathbb{R}^{4 \times 4}$, resulting in:

$$F^{\text{out}} = F^{\text{in}}(W^{\text{bias}} + W^{\cos}\cos\Theta + W^{\sin}\sin\Theta) \tag{5}$$

where $\Theta \in \mathbb{R}$ is a scaler feature of joint, $F^{\text{in}} \in \mathbb{R}^{4 \times 4}$ and $F^{\text{out}} \in \mathbb{R}^{4 \times 4}$ are the input and output link features, corresponding to the parent and child links.

This operator generalizes the classical transformation in Equation 2. When applied recursively, it defines a hierarchical message passing along the robot's kinematic tree. On one hand, in the special case when $\Theta = \theta_j$ and all parameters in Equation 5 are identical to those in Equation 2, this learnable operator degenerate to the forward kinematics of the robot. On the other hand, this generalization also provides a more expressive function space that can be potentially used to encode richer, high-level features beyond joint angles and link poses. These properties enable the operator to combine the inductive bias of kinematic awareness with representational flexibility, making it well-suited for robot learning tasks involving complex actions and motions governed by articulated structures.

### 3.3 Multi-Channel Neural Rodrigues Operator

Derived from the Rodrigues' rotation formula, the neural operator defined in Equation 5 only applies to 1D joint features and $4 \times 4$ link features. We now extend it to a multi-channel operator to learn higher-dimensional features for a single joint. Specifically, we extend the link features from $4 \times 4$ matrices to $F^{\text{in}} \in \mathbb{R}^{C_L \times 4 \times 4}, F^{\text{out}} \in \mathbb{R}^{C'_L \times 4 \times 4}$, and similarly, the joint features $\Theta \in \mathbb{R}^{C_J}$. Here $C_L, C'_L, C_J$ are the channel numbers. Accordingly, the learnable weights are extended to $W^{\text{bias}} \in \mathbb{R}^{C_L \times C'_L \times 4 \times 4}$ and $W^{\cos}, W^{\sin} \in \mathbb{R}^{C_L \times C'_L \times C_J \times 4 \times 4}$, giving a multi-channel extension of Equation 5:

$$U[i, j] = W^{\text{bias}}[i, j] + \sum_{c=1}^{C_J} \left( W^{\cos}[i, j, c]\cos(\Theta[c]) + W^{\sin}[i, j, c]\sin(\Theta[c]) \right) \tag{6}$$

$$F^{\text{out}}[j] = \sum_{i=1}^{C_L} F^{\text{in}}[i]U[i, j], \quad \text{where} \quad U[i, j] \in \mathbb{R}^{4 \times 4} \tag{7}$$

For better expressivity of the network, we further learn another conjugate $\bar{U}[i, j]$ following Equation 6 and extend Equation 7 to include both left and right multiplications:

$$F^{\text{out}}[j] = \sum_{i=1}^{C_L} \left( F^{\text{in}}[i]U[i, j] + \bar{U}[i, j]F^{\text{in}}[i] \right). \tag{8}$$

Equation 6, 8 together define our Multi-Channel Neural Rodrigues Operator with learnable parameters $W^* = \{W^{\text{bias}}, W^{\sin}, W^{\cos}, (\bar{W})^{\text{bias}}, (\bar{W})^{\sin}, (\bar{W})^{\cos}\}$, where $W, \bar{W}$ are the parameters for learning $U, \bar{U}$ respectively. We abbreviate Equation 8 as $F^{\text{out}} = \text{Rodrigues}\left(F^{\text{in}}, W^*, \Theta\right)$ and will use it in the following sections.

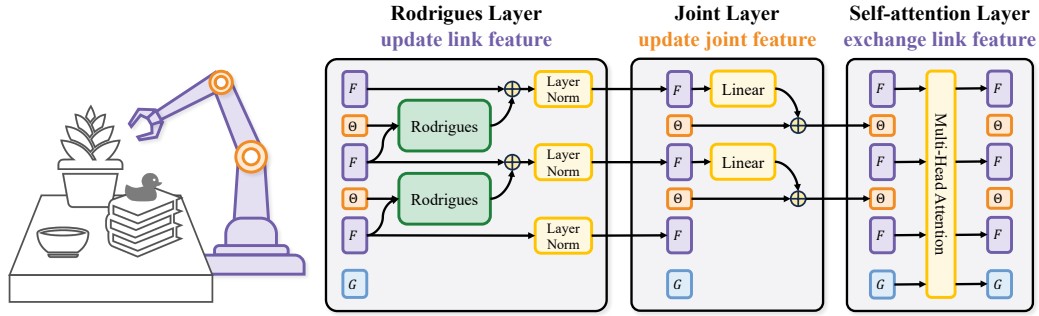

Figure 2: **Rodrigues Block.** It comprises three components: a Rodrigues Layer for passing information from joints to links, constructed with our Multi-Channel Neural Rodrigues Operator; a Joint Layer for passing information from links to joints; and a Self-Attention Layer for global information exchange with all the links and the global token.

## 4 RODRIGUES NETWORK

Given the **Rodrigues Operator**, we are interested in building a complete neural network that leverages this operator while being versatile and expressive. To achieve that, we propose a basic building block called **Rodrigues Block** (Figure 2), which comprises the following three components: (1) a Rodrigues Layer for passing information from joints to links, constructed with our Multi-Channel Neural Rodrigues Operator (Section 4.1); (2) a Joint Layer for passing information from links to joints (Section 4.2); (3) and a Self-Attention Layer (Section 4.3) for global information exchange.

### 4.1 RODRIGUES LAYER

With the Multi-Channel Rodrigues Operator being the core component operating on a single joint, we construct the *Rodrigues Layer*, which extends this operator to the full tree structure of an articulated robot. We define the *feature graph* of an articulated robot as the collection of all link and joint features: $\{F_l\}_{l=0}^D$ for links and $\{\Theta_j\}_{j=1}^D$ for joints. The Rodrigues Layer maintains a set of Rodrigues Kernels $\{W_j^*\}_{j=1}^D$, one for each joint, and computes the output link features $\{F_l^{\text{out}}\}$ from the input link features $\{F_l^{\text{in}}\}$ and joint features $\{\Theta_j^{\text{in}}\}$. For each joint $J_j$, we retrieve its Rodrigues Kernels $W_j^*$, its joint feature $\Theta_j^{\text{in}}$, and the feature of its parent link $F_{\text{p}_j}^{\text{in}}$. We then apply the Multi-Channel Neural Rodrigues Operator (Equation 8) to compute the transformed feature:

$$F_j^{\text{trans}} = \text{Rodrigues}\ \left(F_{\text{p}_j}^{\text{in}}, W_j^*, \Theta_j^{\text{in}}\right) \tag{9}$$

This transformed feature is added to the child link $L_{\text{c}_j}$'s input feature $F_{\text{c}_j}^{\text{in}}$, and normalized:

$$F_{\text{c}_j}^{\text{out}} = \text{LayerNorm}\ \left(F_{\text{c}_j}^{\text{in}} + F_j^{\text{trans}}\right) \tag{10}$$

For the root link, its output feature is defined as its layer-normalized input. This layer updates the link features and leaves the joint features unchanged.

### 4.2 JOINT LAYER

While the Rodrigues Layer updates the link features, we still need a learnable mechanism to update the joint features. The *Joint Layer* computes its output joint features $\{\Theta_j^{\text{out}}\}$ from its own input joint features $\{\Theta_j^{\text{in}}\}$ and link features $\{F_l^{\text{in}}\}$. For each joint $J_j$, we retrieve the feature of its child link $F_{\text{c}_j}^{\text{in}}$, apply a joint-specific linear transformation, and add it to the joint's existing feature:

$$\Theta_j^{\text{out}} = \text{Linear}_j\ \left(\text{Flatten}\ \left(F_{\text{c}_j}^{\text{in}}\right)\right) + \Theta_j^{\text{in}} \tag{11}$$

The transformations $\text{Linear}_j : \mathbb{R}^{C_L \times 4 \times 4} \to \mathbb{R}^{C_J}$ are independently learned for each joint, allowing the model to capture joint-specific information. This layer updates the joint features and leaves the link features unchanged.

### 4.3 OTHER COMPONENTS AND OVERALL ARCHITECTURE

**Self-attention layer** While the Rodrigues Layer and Joint Layer leverage the spatial locality inherent in articulated structures, they restrict information flow to consecutive links and joints. To enable direct communication across all links, we incorporate a *Self-attention Layer*. In this layer, each link feature is first projected into a token using a linear transformation. These tokens then interact through multi-head self-attention, allowing the model to aggregate information from all links regardless of their spatial distance. The attended tokens are subsequently projected back to the link feature space, followed by residual addition and layer normalization. Joint features are left unchanged.

**Global token** To capture and utilize global context, we optionally introduce a global token $G$. This token represents a learnable global feature and only participates in self-attention alongside the link tokens. It is processed through its own projection, residual, and normalization steps, and joins the link tokens during multi-head self-attention. The global token enables the network to store and propagate task-wide information that is not tied to any specific joint or link. We optionally enable the global token in tasks that require predicting global outputs, such as base link pose estimation.

**Overall architecture** We combine the above three components into a unified module called the *Rodrigues Block*. Each Rodrigues Block takes as input the link features, joint features, and an optional global token. It sequentially applies the Rodrigues Layer, Joint Layer, and Self-attention Layer to produce updated link features, joint features, and a global token, where each layer's outputs serves as the next layer's inputs. By stacking multiple such blocks sequentially, we construct the full **Rodrigues Network (RodriNet)**, enabling deep, hierarchical reasoning over articulated structures. The overall architecture is drawn in Figure 2. Refer to Section B of the supplementary for details on computing the first-layer features and task-specific outputs.

## 5 EXPERIMENTS

We evaluate our Rodrigues Network on a set of different tasks, ranging from forward kinematics and motion prediction (Section 5.1), to imitation learning in robotics (Section 5.2), to hand pose estimation for vision and graphics (Section 5.3). The experiments focus on two main questions on action learning: (i) Does the structural prior of articulated kinematics make the Rodrigues Network better at *representing* robot actions? (ii) Can this inductive bias improve the *understanding and prediction* of robot actions in realistic task scenarios? Additional studies on ablations and hyperparameter sensitivity are provided in the supplementary material.

### 5.1 TOY EXPERIMENTS ON KINEMATICS AND MOTION

We begin by studying whether our neural operators help networks better represent the motions and actions of articulated actors with two synthetic tasks: forward kinematics fitting and (Cartesian-space) motion prediction. These synthetic tasks provide a clean environment to directly evaluate network expressivity without other factors.

**Forward kinematics fitting** As discussed in Section 3.1, forward kinematics maps the configuration of an articulated robot (including the root pose and joint angles) to the pose of each link. To effectively control a robot, a neural network should, at a minimum, be capable of learning this mapping. To evaluate whether the Neural Rodrigues Operator possesses strong representational capacity for kinematic modeling, we construct a specialized Rodrigues Network consisting solely of Rodrigues Layers and compare it to other neural backbones in their ability to fit the forward kinematics of a single robot. We conduct this experiment on the LEAP Hand (Shaw et al., 2023), a free-floating dexterous robotic hand with 16 joints and 17 links. The network takes as input the configuration $(T, R, \theta)$ and predicts the pose matrices for all 17 links. For model hyperparameters and training settings, please refer to Sections C.1 and D.1 of the supplementary material.

As shown in Figure 3a, the Rodrigues Network achieves significantly lower prediction error than competing architectures, indicating superior precision in modeling forward kinematics. Moreover, its training loss decreases much faster (Figure 3b), demonstrating better data efficiency. We further visualize the networks' predictions on a single robot configuration in Figure 4. Notably, MLP and

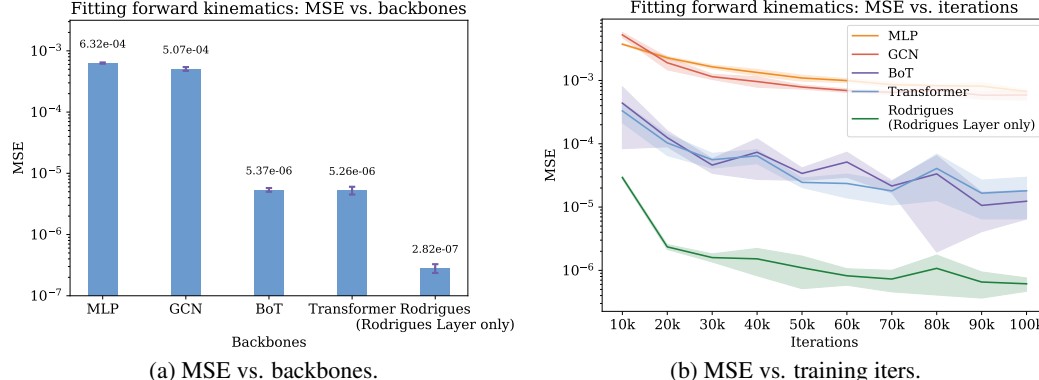

(a) MSE vs. backbones.  (b) MSE vs. training iters.

Figure 3: **Fitting forward kinematics with different network backbones (MSE↓).** The Rodrigues network achieves significantly lower error (left) with faster convergence during training (right).

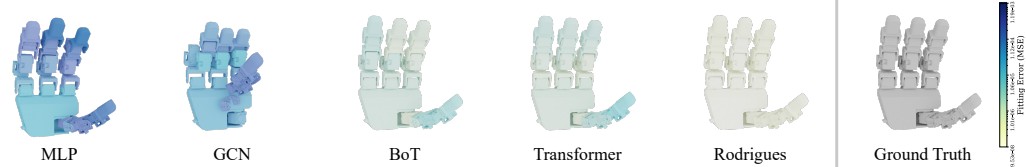

Figure 4: **Visualization of forward kinematics prediction on an example configuration.** Errors are plotted on each link with color scales, with darker colors indicating larger errors.

GCN baselines produce visible artifacts, and all four baseline methods accumulate substantial error near the fingertips. These findings suggest that fitting forward kinematics is not a trivial task; it requires modeling a structured function with spatial and hierarchical dependencies. In contrast, our network closely matches the ground truth across all links, with minimal error, indicating that it effectively captures the underlying structure of the kinematic mapping. We attribute this performance to the inductive bias introduced by the Neural Rodrigues Operator. Its structural prior equips the network with an inherent advantage in learning kinematic functions.

**Motion prediction in Cartesian space**  Although fitting forward kinematics highlights the our operator's representation capacity for kinematics, the task itself has limited practical value. In real-world robot learning scenarios, neural backbones typically process observations in 3D Cartesian space (e.g., point clouds) and output control commands as target joint angles. To evaluate a network's ability to bridge these modalities, we propose a Cartesian motion prediction task that challenges the model to reason about structured motion in Cartesian space while expressing its predictions in joint angle space. This task uses a 6-DoF UR5 robotic arm. We begin by randomly sampling two end-effector poses in 3D space and interpolating between them (linearly for translation and spherically for rotation) to generate a 16-frame Cartesian trajectory. Then, we use inverse kinematics to convert this trajectory into corresponding joint configurations. The network is given the first 8 frames of joint angles and tasked with predicting the remaining 8, also in joint space. Refer to Sections C.2 and D.2 of the supplementary for details on training, model parameter count control, and runtime comparisons.

As shown in Table 1, the Rodrigues Network achieves the lowest training loss and test errors when trained on a pre-collected dataset of $10^5$ trajectories. Notably, its test MSE is lower than the train MSE of all baseline models, indicating that the Rodrigues Network not only fits the data more effectively but also generalizes better without overfitting. Figure 5b further shows that our model consistently outperforms baselines across different training set sizes. In Figure 5a, we visualize the end-effector trajectories predicted by each network for a single example; our model's trajectory aligns most closely with the ground truth. These results confirm that the Rodrigues Network is more effective at bridging Cartesian space and joint angle space than conventional architectures. Please refer to Section C.2 of the supplementary for metric definitions.

Table 1: **Motion prediction in Cartesian space with trainset size $= 10^5$.**

| Backbone | $\text{Error}_T$ (mm) | $\text{Error}_R$ (°) | $\text{Error}_\theta$ (°) | MSE (1e−6) | Train MSE (1e−6) |
|---|---|---|---|---|---|
| MLP | 3.49 ±0.33 | 0.46 ±0.05 | 0.17 ±0.00 | 22.52 ±0.95 | 12.47 ±0.73 |
| GCN | 3.55 ±0.44 | 0.48 ±0.05 | 0.17 ±0.01 | 18.52 ±1.74 | 13.68 ±1.87 |
| BoT | 2.92 ±0.29 | 0.46 ±0.04 | 0.15 ±0.01 | 15.72 ±1.21 | 13.04 ±1.41 |
| Transformer | 2.89 ±0.45 | 0.41 ±0.06 | 0.14 ±0.01 | 12.86 ±1.25 | 10.50 ±1.21 |
| Rodrigues | **1.21** ±**0.17** | **0.16** ±**0.04** | **0.06** ±**0.00** | **2.56** ±**0.39** | **1.93** ±**0.34** |

(a) **Trajectory visualization.** We visualize the trajectories of the end-point (marked in red) predicted by each model from the top-down view, interpolated with B-spline curves.

(b) Testset performance (MSE↓) under different amounts of training data.

Figure 5: **Results for motion prediction in Cartesian space.**

## 5.2 ROBOTIC MANIPULATION WITH IMITATION LEARNING

Next, we evaluate whether our method benefits realistic robotic applications. We integrate the Rodrigues Network as a backbone into the Diffusion Policy (Chi et al., 2023), one of the state-of-the-art imitation learning frameworks, and test on a manipulation benchmark in simulation.

**Benchmark** We construct a suite of five manipulation tasks from ManiSkill (Mu et al., 2021) using a 7-DoF Franka arm with a 1-DoF Panda gripper, simulated in SAPIEN (Xiang et al., 2020). For each task, we collect 100–500 demonstration trajectories using motion planning, with each trajectory spanning 200 steps. During training, the network receives proprioception and object state as input and outputs relative joint offsets used for PD control. Performance is measured by running 100 evaluation rollouts in simulation, and all models are trained with 5 random seeds to report the mean and standard deviation of success rates.

**Methods** We adopt Diffusion Policy (Chi et al., 2023) as our learning framework, using a 2-frame observation history and predicting 16 future steps, of which 8 are executed during deployment. To isolate the impact of the neural backbone, we keep the outer framework and all other components fixed, modifying only the denoising network. Baseline architectures include the U-Net and Transformer designs from the original Diffusion Policy paper. Our method replaces these with the Rodrigues Network, which takes the current observation, denoising timestep, and a noisy action as inputs and predicts the corresponding action noise. The gripper output is handled via the global token. All models are tuned to have approximately 17 million parameters for a fair comparison. Implementation details are provided in Section D.3 of the supplementary material.

**Results and analysis** As shown in Table 2, Diffusion Policy (Chi et al., 2023) with the Rodrigues Network backbone achieves overall state-of-the-art performance, demonstrating that our kinematics-inspired inductive bias improves imitation learning in realistic robotic tasks. In particular, we observe substantial gains in *PickCube* and *StackCube*, and comparable or slightly better performance in *PushCube*, *PegInsertionSide*, and *PlugCharger*. These results suggest that the benefits of the Rodrigues Decoder are task-dependent. *PushCube* is relatively simple, leading all backbones to achieve near-perfect success. In contrast, *PegInsertionSide* and *PlugCharger* involve complex contact dynamics, such as inserting a peg into a hole, where tactile or force feedback would be beneficial. However, these sensor inputs are not provided in the environment, making the neural backbone less of a limiting factor. In summary, enhancing the backbone architecture with our neural operator can yield significant gains when the network is the performance bottleneck.

Table 2: **Baseline comparisons on the imitation learning benchmark.** Simulated success rate.

| Method | PushCube | PickCube | StackCube | PegInsertionSide | PlugCharger | Average |
|---|---|---|---|---|---|---|
| Transformer-DP | 0.98 ±0.02 | 0.63 ±0.05 | 0.38 ±0.02 | 0.18 ±0.05 | 0.04 ±0.02 | 0.44 |
| UNet-DP | **1.00** ±0.00 | 0.85 ±0.03 | 0.37 ±0.04 | 0.56 ±0.06 | **0.13** ±0.06 | 0.58 |
| Rodrigues-DP | **1.00** ±0.00 | **0.94** ±0.02 | **0.44** ±0.05 | **0.58** ±0.04 | 0.10 ±0.02 | **0.61** |

Table 3: **Baseline comparisons on the FreiHAND dataset.** We use the standard protocol and report metrics on 3D joint and 3D mesh accuracy. PA-MPVPE and PA-MPJPE numbers are in mm.

| Method | PA-MPJPE $\downarrow$ | PA-MPVPE $\downarrow$ | F@5 $\uparrow$ | F@15 $\uparrow$ |
|---|---|---|---|---|
| I2L-MeshNet (Moon & Lee, 2020) | 7.4 | 7.6 | 0.681 | 0.973 |
| Pose2Mesh (Choi et al., 2020) | 7.7 | 7.8 | 0.674 | 0.969 |
| I2UV-HandNet (Chen et al., 2021a) | 6.7 | 6.9 | 0.707 | 0.977 |
| METRO (Lin et al., 2021a) | 6.5 | 6.3 | 0.731 | 0.984 |
| Tang *et al.* (Tang et al., 2021) | 6.7 | 6.7 | 0.724 | 0.981 |
| Mesh Graphormer (Lin et al., 2021b) | 5.9 | 6.0 | 0.764 | 0.986 |
| MobRecon (Chen et al., 2022) | **5.7** | 5.8 | 0.784 | 0.986 |
| AMVUR (Jiang et al., 2023) | 6.2 | 6.1 | 0.767 | 0.987 |
| HaMeR (Pavlakos et al., 2024) | 6.0 | 5.7 | 0.785 | 0.990 |
| HaMeR (Reproduced) | 6.2 | 5.9 | 0.774 | 0.989 |
| **Ours** | 5.9 | **5.6** | **0.793** | **0.991** |

## 5.3 3D HAND RECONSTRUCTION

Finally, we show that Rodrigues Networks extend beyond robots to animated characters. Specifically, we apply our method to 3D hand reconstruction from single-view RGB images, which involves predicting the rotations and positions of hand joints based on the kinematic structure defined by the MANO (Romero et al., 2022) hand kinematics.

Our network builds upon HaMeR (Pavlakos et al., 2024) by replacing its vanilla transformer with the proposed Rodrigues Network (with modifications to suit MANO's configuration representation). Additionally, we introduce a cross-attention layer following the self-attention layer to enable interactions between joint and link features and the input image tokens. Our network achieves a notable performance improvement while significantly reducing the number of parameters (39.5M vs. ours: **10.7M**). Further architectural details are provided in Section A of the supplementary material.

Table 3 presents the quantitative results. We follow the evaluation protocol and metrics established by HaMeR (Pavlakos et al., 2024), and report performance on the FreiHand (Zimmermann et al., 2019) dataset. For reference, results of our reproduced HaMeR model are also included.

As shown in Table 3, our method achieves superior quantitative performance, surpassing the previous state-of-the-art. Compared to the strongest baseline, HaMeR, our approach outperforms both the results reported in the original paper and our reproduced implementation. This underscores the effectiveness of incorporating hand joint kinematics into the network, resulting in substantial improvements on kinematics-related tasks. Therefore, our approach is not limited to robotic applications, demonstrating its versatility and applicability to graphics-related tasks as well.

## 6 CONCLUSIONS AND DISCUSSIONS

In this work, we design a neural network that addresses the kinematic structural priors in articulated robot action learning. Core to it is the Neural Rodrigues Operator that extends the Rodrigues' rotation formula into a learnable operator of more flexible forms, providing networks with an inductive bias that can better model kinematics-related features. With this neural operator as a key component, we build the Rodrigues Network with additional layer designs, resulting in a powerful and embodiment-aware network architecture applicable to diverse action-learning tasks.

Many state-of-the-art networks in robot learning build upon architectural designs originally developed for other domains, such as vision and language. With this work, we aim to encourage the

exploration of action-centric neural network architectures that are tailored to the unique characteristics of robot learning, particularly those aspects that are underexplored in other fields.

**Limitations and future work**    First, while Rodrigues Networks successfully incorporate articulated kinematics as an inductive bias, they do not yet account for the geometry of individual links. In many settings, this information is available and could improve performance on tasks that require precise contact reasoning. Second, the current Neural Rodrigues Operator is restricted to rotational joints. Extending it to also handle translational joints would broaden its applicability. Third, our robot learning experiments focused on imitation learning. Exploring reinforcement learning (RL) scenarios could further test the network's generality and effectiveness in closed-loop settings.

## 7    ACKNOWLEDGEMENTS

Leonidas Guibas, Yang You and Congyue Deng acknowledge support from the Toyota Research Institute University 2.0 Program, ARL grant W911NF-21-2-0104, a Vannevar Bush Faculty Fellowship, and a gift from the Flexiv corporation. Haoran Geng acknowledges support from the Berkeley Fellowship Award and the Qualcomm Innovation Fellowship. Yang You is also supported in part by the Outstanding Doctoral Graduates Development Scholarship of Shanghai Jiao Tong University. Congyue Deng is also supported in part by the Tayebati Postdoctoral Fellowship at the MIT Schwarzman College of Computing. Pieter Abbeel holds concurrent appointments as a professor at UC Berkeley and as an Amazon Scholar, and this paper is not associated with Amazon.

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

## Supplementary Material

This supplementary material provides additional details to support the experiments presented in the main paper. Section A describes how we adapt the Neural Rodrigues Operator for animated characters, as used in the 3D hand reconstruction experiment. Section B provides additional details on network architecture. Section C outlines our experimental settings, including data collection, training procedures, evaluation protocols, and all training hyperparameters. Section D details the architecture and configurations of both baseline models and our proposed method across all experiments. These details are provided to facilitate reproducibility of our results. We will release our code, datasets, and pretrained checkpoints upon acceptance to further support the community in reproducing and building upon our work.

Additionally, Section E presents further experiments designed to investigate the properties of our model and provide additional support for the claims made in the main paper. Finally, Section F describes our CUDA implementation of the Multi-Channel Rodrigues Operator, which accelerates training and inference.

## A    Rodrigues Network for animated characters

Beyond articulated robots, animated characters such as MANO (Romero et al., 2022) and SMPL (Loper et al., 2023) also exhibit articulated structures and employ forward kinematics, which can similarly serve as inductive biases for neural networks. However, a fundamental distinction exists between these models and articulated robots: each joint in MANO and SMPL is a free rotational joint with three degrees of freedom, as opposed to the single-axis, 1-DoF rotational joints typically found in robots. As a result, adapting our approach to animated characters requires a fundamental modification of the Rodrigues Operator to accommodate unconstrained 3-DoF rotations. We need to identify the basic operation in forward kinematics for animated characters, and develop a neural operator by turning that operation learnable.

**Forward kinematics**    At each 3-DoF joint, we can parameterize its configuration as a unit quaternion $\mathbf{q} = (q_w, q_i, q_j, q_k)$ where $\|\mathbf{q}\| = 1$. Following this, forward kinematics for animated characters is similar to articulated robots. There are also two transformations at each joint $J_j$: (i) a fixed coordinate change $\mathbf{T}_j$ from the parent link frame to the joint frame; (ii) a dynamic rotation $\mathbf{R}_{\text{q2mat}}(\mathbf{q}) \in \mathbb{R}^{3\times3}$ in the joint frame. The parent-to-children pose transformation can also be written as:

$$\begin{bmatrix} \mathbf{R}_{\mathrm{c}_j} & \mathbf{t}_{\mathrm{c}_j} \\ \mathbf{0}_{1\times3} & 1 \end{bmatrix} = \begin{bmatrix} \mathbf{R}_{\mathrm{p}_j} & \mathbf{t}_{\mathrm{p}_j} \\ \mathbf{0}_{1\times3} & 1 \end{bmatrix} \mathbf{T}_j \begin{bmatrix} \mathbf{R}_{\text{q2mat}}(\mathbf{q}) & \mathbf{0}_{3\times1} \\ \mathbf{0}_{1\times3} & 1 \end{bmatrix}. \tag{12}$$

Forward kinematics for animated characters is essentially a hierarchical combination of Equation 12.

**Quaternions to matrix conversion**    The key difference of computing forward kinematics for animated characters lies in converting a joint's unit quaternions into a 3 by 3 rotation matrix. This can be done using the following formula:

$$\mathbf{R}_{\text{q2mat}}(\mathbf{q}) = \begin{bmatrix} 1 - 2(q_j^2 + q_k^2) & 2(q_i q_j - q_k q_w) & 2(q_i q_k + q_j q_w) \\ 2(q_i q_j + q_k q_w) & 1 - 2(q_i^2 + q_k^2) & 2(q_j q_k - q_i q_w) \\ 2(q_i q_k - q_j q_w) & 2(q_j q_k + q_i q_w) & 1 - 2(q_i^2 + q_j^2) \end{bmatrix} \tag{13}$$

**Neural Rodrigues Operator for quaternions**    Observe from Equation 13 every entry in $\mathbf{R}_{\text{q2mat}}(\mathbf{q})$ is a linear combination of ones and all 10 quadratic terms of $q_w, q_i, q_j, q_k$, with constant coefficients. Thus, we can mimic what we have done for the Rodrigues' rotation formula and re-parameterize Equation 12 as:

$$\mathbf{P}_{\mathrm{c}_j} = \mathbf{P}_{\mathrm{p}_j} \left( \mathbf{A} + \sum_{x,y \in \{w,i,j,k\}, x \neq y} \mathbf{A}_{xy} q_x q_y + \sum_{x \in \{w,i,j,k\}} \mathbf{A}_{xx} q_x^2 \right) \tag{14}$$

where $\mathbf{A}, \mathbf{A}_{xy}, \mathbf{A}_{xx} \in \mathbb{R}^{4\times4}$ are coefficient matrices that only depend on the joint's structure. Similar to what we have done for axis angles, we construct our Neural Rodrigues Operator for

quaternions by replacing these fixed coefficient with learnable weights $W^{\text{bias}}, W^{xy}, W^{xx} \in \mathbb{R}^{4 \times 4}$, resulting in:

$$F^{\text{out}} = F^{\text{in}} \left( W^{\text{bias}} + \sum_{x,y \in \{w,i,j,k\}, x \neq y} W^{xy} q_x q_y + \sum_{x \in \{w,i,j,k\}} W^{xx} q_x^2 \right) \quad (15)$$

where $F^{\text{in}} \in \mathbb{R}^{4 \times 4}$ and $F^{\text{out}} \in \mathbb{R}^{4 \times 4}$ are the input and output link features, corresponding to the parent and child links. Similar to the original Neural Rodrigues Operator, we also extend this operator to support multi link feature channels and left multiplication. But since there are already four feature channels for each joint, we do not extend it further. We also add learnable weights $W^x \in \mathbb{R}^{4 \times 4}$ to Equation 15 to support representing linear terms of $q_w, q_i, q_j, q_k$.

## B  ADDITIONAL DETAILS ON NETWORK ARCHITECTURE

This section describes how task observations are mapped to the Rodrigues Network's inputs and how the network's outputs are mapped to task actions.

**Input embeddings**  In all experiments except 3D hand reconstruction, task inputs are low-dimensional vectors. We use a linear input embedding layer to map the input observation feature to the Rodrigues Network's input. Specifically, given an observation feature of dimension $d_{\text{obs}}$, we apply separate linear transformations to produce the initial link features, joint features, and, if applicable, a global token. These serve as inputs to the first Rodrigues Block of the Rodrigues Network. After processing through the stacked Rodrigues Blocks, we obtain output features for all links, joints, and the global token.

For 3D hand reconstruction, the inputs are RGB images. We first apply a ViT encoder to transform the image into visual tokens, then insert a cross-attention layer into each Rodrigues Block so that link features and the global token can attend to these tokens. As for the input to the first Rodrigues Block in this case, learnable embeddings are used.

**Output heads**  For joint-level outputs, each joint's final output feature is concatenated with the output feature of its child link, then passed through a joint-specific linear transformation to produce the final output. For global outputs (those not associated with a specific joint), we apply a separate linear transformation to the final global token.

## C  EXPERIMENT SETTINGS

### C.1  FORWARD KINEMATICS FITTING

**Data preparation**  In this experiment, the models learn to fit the forward kinematics mapping of a robot. Therefore, we do not need to generate a fixed-size training set. However, we do create a fixed-size validation set for selecting the best checkpoint, and a separate test set for evaluation. We conduct this experiment on the LEAP hand, which is a fully-actuated robotic hand with four fingers, 16 rotational joints, and 17 links. Each data point includes the input: the root link position $T \in \mathbb{R}^3$, the root link orientation matrix $R \in \mathbb{R}^{3 \times 3}$, and joint angles $\theta \in \mathbb{R}^{16}$. The output is the position vectors and orientation matrices of all 17 links. We sample $T$ uniformly in $[-0.05 \text{ cm}, 0.05 \text{ cm}]^3$, $R$ uniformly from $\text{SO}(3)$, and $\theta$ uniformly within the joint limits. We then compute all link poses using forward kinematics. The validation and test sets each contain 10,000 input-output pairs, generated using different random seeds.

**Training details**  We use the Adam optimizer with $\text{lr} = 0.0003$ and without weight decay and train each model for 10,000 steps. In each training step, we generate a new batch of 1024 input-output pairs using the same method as for the validation and test sets, instead of sampling from a fixed-size training set. The model takes the input and predicts 17 position vectors and 17 orientation matrices, totaling $17 \times (3 + 9)$ values. We compute the mean squared error (MSE) loss between the predicted and ground truth outputs. The loss is averaged over the batch and all output values, and then used to update the model through gradient descent. We evaluate the model on the validation

set for every 500 steps, and pick the model with the lowest validation loss for final evaluation. We also save a checkpoint every 1,000 iterations for plotting the training curve. All hyperparameters are summarized in Table 4.

Table 4: **Training hyperparameters for forward kinematics fitting experiment.**

| Parameter name | Value | Parameter name | Value | Parameter | Value |
|---|---|---|---|---|---|
| Training iterations | 100,000 | Optimizer | Adam | Learning rate | 0.0003 |
| Batch size | 1024 | Validate every | 500 iterations | Weight decay | 0 |

## C.2 Motion prediction in Cartesian space

**Data preparation** This experiment is conducted on a fixed-base 6-DoF UR5 robotic arm. To make the inverse kinematics solution unique for all reachable end-effector poses, we limit the six joint ranges to $[0, \pi/2]$, $[-\pi/2, 0]$, $[0, \pi/2]$, $[0, \pi/4]$, $[0, \pi/4]$, and $[0, \pi/4]$. We sample two joint configurations $\boldsymbol{\theta}_{\text{start}}, \boldsymbol{\theta}_{\text{end}} \in \mathbb{R}^6$ within these limits, which serve as the start and end poses. We then use forward kinematics to compute the corresponding 3D end-effector poses $\mathbf{P}_{\text{start}}, \mathbf{P}_{\text{end}} \in \text{SE}(3)$. Next, we interpolate $16 - 2$ intermediate 3D poses between the start and end to form a trajectory of 16 poses: $\mathbf{P}_1, \cdots, \mathbf{P}_{16}$. Positions are interpolated linearly, and orientations are interpolated using spherical linear interpolation (slerp). Finally, we apply inverse kinematics to get the corresponding joint configurations $\boldsymbol{\theta}_1, \cdots, \boldsymbol{\theta}_{16}$ for the 3D pose trajectory. The first 8 frames of joint configurations are used as input, and the models are asked to predict the last 8 frames. In Cartesian space, the motion pattern is clear: positions follow a straight line, and orientations change smoothly with slerp. However, this pattern is not obvious in joint space. The model needs to learn to reason about this motion pattern from the joint configurations. We generated four training sets with $10^3$, $10^4$, $10^5$, and $10^6$ input-output pairs. The validation and test sets each contain $10^4$ input-output pairs.

**Training details** We use the Adam optimizer with a learning rate of 0.0001 and no weight decay, and train each model for 10,000 steps. Each training session uses a fixed training set. In each step, we randomly sample a batch of 1,024 input-output pairs from this set. The model takes the input and predicts $8 \times 6$ values, which represent the joint configurations for the last 8 frames. We compute the mean squared error (MSE) loss between the predicted and ground truth outputs, and use it to perform gradient descent. We evaluate the model on the validation set every 500 steps, and select the checkpoint with the lowest validation loss for final testing. Unlike the forward kinematics fitting experiment, this experiment trains models on a fixed-size training set and evaluates them on a separate test set drawn from the same distribution. Therefore, it evaluates not only the model's ability to fit the training data, but also its generalization performance. All training hyperparameters are listed in Table 5.

Table 5: **Training hyperparameters for motion prediction in Cartesian space experiment.**

| Parameter name | Value | Parameter name | Value | Parameter | Value |
|---|---|---|---|---|---|
| Training iterations | 100,000 | Optimizer | Adam | Learning rate | 0.0001 |
| Batch size | 1024 | Validate every | 500 iterations | Weight decay | 0 |
| Input frames | 8 | Output frames | 8 | DoFs | 6 |

**Evaluation** We apply forward kinematics to the predicted joint configurations to obtain the predicted end-effector 3D poses. Then, we compare both the predicted joint configurations and the predicted end-effector poses with the ground truth. We report the following evaluation metrics: 1) $\text{Error}_T$ (mm): End-effector position error on the test set, averaged over the 8 predicted frames. 2) $\text{Error}_R$ (°): End-effector orientation error on the test set, averaged over the 8 predicted frames. We compute the relative rotation between the predicted and ground truth orientations, convert it to axis-angle representation, and take the angle. 3) $\text{Error}_\theta$ (°): Absolute joint configuration error on the test set, averaged over the 8 predicted frames and 6 joints. 4) MSE ($\times 10^{-6}$): Mean squared error on the test set, multiplied by $10^{-6}$ for better readability. 5) Train MSE ($\times 10^{-6}$): Mean squared error on the training set, also scaled by $10^{-6}$. The first four metrics measure the model's generalization ability, while the last one reflects its fitting ability. For all metrics, lower values indicate better performance.

### C.3 ROBOTIC MANIPULATION WITH IMITATION LEARNING

**Data generation**   We evaluate on five representative manipulation tasks from the ManiSkill Benchmark (Mu et al., 2021): PushCube, PickCube, StackCube, PegInsertionSide, and PlugCharger. For each task, we collect expert demonstration trajectories using the ManiSkill data collection API. These trajectories are generated via motion planning algorithms and are limited to a maximum of 200 simulation steps per episode. The number of demonstrations collected per task is listed in Table 7. During testing, the initial scene configurations are randomized using the same distribution employed during training data collection. For additional details, please refer to the ManiSkill benchmark (Mu et al., 2021).

**Training details**   We follow the training setup in Chi et al. (2023), including optimizer choice, learning rate scheduling, and exponential moving average (EMA) for network stabilization. Specifically, we use the AdamW optimizer with a learning rate of $0.0001$, $\beta = (0.95, 0.999)$, and a weight decay of $1 \times 10^{-6}$. A cosine learning rate scheduler with 500 warm-up steps is applied. To enhance training speed and stability, we maintain an EMA of the model weights with a decay factor of 0.75. During training, we sample a batch of 1024 expert observation-action pairs from the demonstration dataset in each iteration and perform a gradient update using the diffusion loss. All training hyperparameters are summarized in Table 6. The number of training iterations and demonstration trajectories varies across tasks, depending on task complexity; see Table 7 for details. For further implementation details, please refer to Chi et al. (2023).

Table 6: **Training hyperparameters for imitation learning experiment (following Chi et al. (2023)'s settings).**

| Parameter name | Value | Parameter name | Value | Parameter | Value |
|---|---|---|---|---|---|
| Optimizer | AdamW | Learning rate | 0.0001 | Weight decay | 1e-6 |
| LR scheduler | Cosine scheduler | Batch size | 1024 | Episode steps | 200 |

Table 7: **Demo trajectories and training iterations for each task in imitation learning experiment.**

| Task name | PushCube | PickCube | StackCube | PegInsertionSide | PlugCharger |
|---|---|---|---|---|---|
| Demo trajectories | 100 | 100 | 100 | 500 | 500 |
| Training iterations | 30k | 30k | 60k | 100k | 300k |

### C.4 3D HAND RECONSTRUCTION

**Dataset preparation**   Following Hamer, we train our model on a large number of datasets that provide 2D or 3D hand annotations. Specifically, we use FreiHAND (Zimmermann et al., 2019), HO3D (Hampali et al., 2020), MTC (Xiang et al., 2019), RHD (Zimmermann & Brox, 2017), InterHand2.6M (Moon et al., 2020), H2O3D (Hampali et al., 2020), DEX YCB (Chao et al., 2021), COCO WholeBody (Jin et al., 2020), Halpe (Fang et al., 2022) and MPII NZSL (Simon et al., 2017).

**Training Details**   We follow the training protocol described in HaMeR (Pavlakos et al., 2024). We use the AdamW optimizer with learning rate 1e-5 and weight decay 1e-4, and train the model for 1,000,000 steps. The batch size is set to 64. Our model takes in a square image around the target hand, resized to 256 x 256, and output the 58 MANO (Romero et al., 2022) parameters of the human hand, specifically 48 pose parameters and the 10 shape parameters. We evaluate the model on the validation set for every 1,000 steps, and pick the model with the lowest validation loss for final evaluation. Similar to Hamer, we additionally estimate the camera parameters $\pi$. The camara $\pi$ corresponds to a translation $t \in \mathbb{R}^3$ that projects the 3D mesh and joints into the image. Hyperparameters are summarized in Table 8.

Table 8: **Training hyperparameters for 3D hand reconstruction experiment.**

| Parameter | Value | Parameter | Value | Parameter | Value |
|---|---|---|---|---|---|
| Training iterations | 1,000,000 | Optimizer | AdamW | Learning rate | 1e-5 |
| Batch size | 64 | Validate every | 1,000 iterations | Weight decay | 1e-4 |

## D  METHOD AND IMPLEMENTATION DETAILS

### D.1  FORWARD KINEMATICS FITTING

We construct four baseline methods using existing neural network backbones for comparison: MLP, Graph Convolution Network (GCN), Transformer, and Body Transformer (BoT) (Sferrazza et al., 2024).

**MLP**  The MLP baseline concatenates the input root position, root orientation, and joint angles into a $(3 + 9 + \text{DoF})$-dimensional vector ($\text{DoF} = 16$ for the LEAP hand), and feeds it into a 7-layer MLP with the following shape: $[28, 768, 768, 768, 768, 768, 768, 204]$. The output represents the positions and orientations of all $\text{DoF} + 1 = 17$ links. All hidden layers use ReLU activation, and no normalization layers are applied.

**GCN**  The GCN baseline uses $1 + \text{DoF}$ separate linear transformations to encode the root pose and each joint angle into $1 + \text{DoF}$ feature embeddings, each with 512 dimensions. Each embedding corresponds to one robot link: the one derived from the root pose represents the root link, and each joint's embedding corresponds to its child link. We represent the robot as an undirected tree graph with $1 + \text{DoF}$ nodes (links) and $\text{DoF}$ edges (joints). The GCN applies 11 layers of graph convolution to update the link features. All hidden layers have 512 dimensions. Finally, separate linear transformations are used to predict the pose (position and orientation) of each link.

**Transformer**  The Transformer baseline uses the same approach as the GCN baseline to encode the input root pose and joint angles into $1 + \text{DoF}$ link feature embeddings, each with 256 dimensions. After applying positional encoding, these embeddings are passed through a Transformer backbone consisting of 8 Transformer blocks. Each block includes a feed-forward layer with a 256-dimensional hidden layer. The Transformer outputs $1 + \text{DoF}$ updated link features. As in the GCN baseline, separate linear transformations are used to predict the pose of each link from these features.

**Body Transformer (BoT) (Sferrazza et al., 2024)**  The BoT baseline shares a similar architecture with the Transformer baseline but replaces the Transformer backbone with a Body Transformer backbone. It uses the same robot connectivity graph as in the GCN baseline to capture the structural relationships between links.

**Rodrigues Network (ours)**  Our method uses a specially customized Rodrigues Network as the neural backbone. The network consists of 12 Rodrigues Blocks, and each block contains only a single Rodrigues Layer. No Joint Layers, Self-Attention Layers, or Global Tokens are used. Each Rodrigues Layer has 1 joint channel ($C_J = 1$) and 3 link channels ($C_L = 3$). We first concatenate the input root position, root orientation, and joint angles into a $(3 + 9 + \text{DoF})$-dimensional vector. This vector is passed through DoF separate linear transformations to produce the input joint features, and through $1 + \text{DoF}$ separate linear transformations to produce the input link features. The Rodrigues Network updates the joint and link features through its 12 Rodrigues Blocks, resulting in the output joint and link features. Finally, we concatenate all output link features and apply $1 + \text{DoF}$ separate linear transformations to predict the pose of each link.

**Parameter count comparison**  All baseline networks have around 3 million parameters, while our network has only 0.2 million parameters. Since this experiment focuses only on fitting ability and not generalization, this setup gives an advantage to the baseline methods with more parameters. Even so, our network still performs better.

**Compute resources** We train each model on a single Quadro RTX 6000 graphics card, and the approximate training time for 100,000 training iterations are listed in Table 9.

Table 9: **Approximate training time of different methods for fitting forward kinematics**

| Method | MLP | GCN | Transformer | BoT | Rodrigues (ours) |
|--------|-----|-----|-------------|-----|------------------|
| Time | 17min | 1h 50min | 2h 20min | 2h 20min | 1h 18min |

## D.2 MOTION PREDICTION IN CARTESIAN SPACE

This experiment uses the same four neural backbones as in the forward kinematics fitting experiment: MLP, Graph Convolution Network (GCN), Transformer, and Body Transformer (BoT).

**MLP** The MLP baseline takes all $8 \times 6 = 48$ input values from the 8 input frames as input and feeds them into a 7-layer MLP with the following layer sizes: $[48, 768, 768, 768, 768, 768, 768, 48]$. The output represents the joint configurations of the 8 predicted frames. All hidden layers use ReLU activation, and no normalization layers are applied.

**GCN** The GCN baseline uses $1 + \text{DoF}$ separate linear transformations to encode the input into $1 + \text{DoF}$ feature embeddings, each with 512 dimensions. As in the forward kinematics fitting experiment, each embedding corresponds to one robot link, and the robot is modeled as an undirected tree graph with $1 + \text{DoF}$ nodes (links) and DoF edges (joints). The GCN applies 11 layers of graph convolution to update the link features. Finally, for each joint, we extract the output feature of its corresponding child link and apply a separate linear transformation to predict the joint's angles for all 8 output frames.

**Transformer** The Transformer baseline follows the same procedure as the GCN baseline to encode the input into $1+\text{DoF}$ link feature embeddings, each with 250 dimensions. Positional encoding is then applied to these embeddings, which are processed by a Transformer backbone consisting of 8 Transformer blocks. Each block contains a feed-forward layer with a 250-dimensional hidden layer. The Transformer outputs $1 + \text{DoF}$ updated link features. As in the GCN baseline, we extract the output feature corresponding to each joint's child link and apply a separate linear transformation to predict the joint's 8 output angles.

**Body Transformer (BoT) (Sferrazza et al., 2024)** The BoT baseline replaces the Transformer backbone of the Transformer baseline with a Body Transformer backbone, using the same number of blocks and hidden dimensions.

**Rodrigues Network (ours)** Our method uses a full Rodrigues Network as the neural backbone, consisting of 12 Rodrigues Blocks. Each block contains a Rodrigues Layer, a Joint Layer, and a Self-attention Layer. In this experiment, Global Tokens are also not used. Each Rodrigues Layer is configured with 4 joint channels ($C_J = 4$) and 8 link channels ($C_L = 8$). Each Self-attention Layer operates on 256-dimensional embeddings with 8 attention heads. To prepare the input, we apply DoF separate linear transformations to produce the input joint features, and $1 + \text{DoF}$ separate linear transformations to generate the input link features. These features are then processed by the Rodrigues Network, which updates them through all 12 blocks to produce the final joint and link features. For each joint, we concatenate its output joint feature with the output link feature of its child link, and apply a separate linear transformation to predict the joint's 8-frame output angles.

**Parameter count comparison** All baseline methods and our approach have approximately 3 million parameters. This ensures a fair comparison of both fitting and generalization abilities.

**Compute resources** We train each model on a single Quadro RTX 6000 graphics card, and the approximate training time for 100,000 training iterations are listed in Table 10.

Table 10: **Approximate training time of different methods for motion prediction experiment**

| Method | MLP | GCN | Transformer | BoT | Rodrigues (ours) |
|---|---|---|---|---|---|
| Time | 27min | 1h 12min | 1h 20min | 1h 20min | 2h 22min |

## D.3 ROBOTIC MANIPULATION WITH IMITATION LEARNING

This experiment follows the Diffusion Policy (DP) framework proposed in Chi et al. (2023), and evaluates different neural backbones for denoising action samples. The original paper (Chi et al., 2023) provides two backbone options, convolutional UNet and Transformer, which we directly adopt as baselines. In the DP framework, the policy observes a history of 2 prior time steps, predicts a sequence of 16 future actions, and executes the first 8 during deployment. Each action vector consists of 8 values: 7 for joint position control of the 7-DoF Franka arm and 1 for the 1-DoF Panda gripper.

**UNet-DP**   The UNet-DP baseline uses a convolutional UNet architecture that performs temporal convolution over the 16-step action sequence. We follow the default structure, with the hidden dimensions of the downsampling layers set to $[320, 320, 344]$. For further implementation details, please refer to Chi et al. (2023).

**Transformer-DP**   The Transformer-DP baseline applies causal self-attention to the 16 action-step tokens. We use an embedding dimension of 320, feed-forward hidden dimension of 1280, and a total of 10 Transformer blocks. Additional details are available in Chi et al. (2023).

**Rodrigues-DP (ours)**   Our method adopts a Rodrigues Network as the denoising backbone, composed of 12 Rodrigues Blocks with 16 link feature channels ($C_L = 16$), 8 joint feature channels ($C_J = 8$), and a self-attention embedding dimension of 256. We also introduce Global Tokens of dimension 128 to model gripper actions. The denoising network takes three inputs: a time step, a noisy action, and an observation vector. We first embed the time step and concatenate it with the observation and noisy action to form an input feature vector. This vector is then passed through DoF separate linear transformations to generate the input joint features, $1 + \text{DoF}$ separate linear transformations to generate the input link features, and one additional linear transformation to produce the input global token. The Rodrigues Network updates all features through its 12 Rodrigues Blocks, resulting in output joint features, output link features, and output global token. For each joint, we concatenate its output joint feature with the output link feature of its corresponding child link, and apply a separate linear transformation to predict the joint's 16-frame denoised action trajectory. For the gripper, we use a linear transformation on the output global token to predict the 16-frame denoised gripper actions.

**Parameter count comparison**   All baseline networks and our network have approximately 17 million parameters to make comparisons fair.

## D.4 3D HAND RECONSTRUCTION

Architecture-wise, we employ a Vision Transformer (Dosovitskiy et al., 2020) as the image-processing backbone, followed by our Rodrigues Network (RodriNet) head to regress both hand and camera parameters. Since there are 17 links and 16 joints in the MANO parameter model, to make the information pass through the entire kinematic chain, our RodriNet head comprises 18 Rodrigues blocks, each of which sequentially applies a Rodrigues Layer, a Joint Layer, and a Self-attention Layer to update the link features, joint features. The Rodrigues Layer uses 4 link channels, and the Self-attention Layer uses 64 embed dimensions. We enable Global Tokens in this experiment to model the MANO's shape parameters.

In addition to the standard components, we append a Cross-Attention Layer at the end of each block. This layer processes the link features and the input image tokens through a standard cross-attention transformer module, enriching the link features with visual information from the input image.

The training takes approximately 7 days on a single Quadro RTX 6000 graphics card.

Table 11: **Ablation studies for motion prediction in Cartesian space with trainset size $= 10^5$.** We remove the Rodrigues Layer (R Layer), Joint Layer (J Layer), or Self-attention Layer (S Layer) respectively from the original Rodrigues Network, and evaluate the MSE on train/test sets.

| R Layer | J Layer | S Layer | Params (M) | Train MSE (1e$-$6) | Test MSE (1e$-$6) |
|---------|---------|---------|------------|--------------------|-------------------|
| ✓ | ✓ | ✓ | 3.04 | 1.93 $\pm 0.34$ | 2.56 $\pm 0.39$ |
| ✓ | ✓ |   | 1.44 | 1.94 $\pm 0.26$ | 2.33 $\pm 0.26$ |
| ✓ |   | ✓ | 3.01 | 2.33 $\pm 0.56$ | 2.80 $\pm 0.62$ |
|   | ✓ | ✓ | 1.69 | 5.57 $\pm 0.55$ | 6.19 $\pm 0.57$ |

## E  ADDITIONAL RESULTS

We conduct additional experiments to further support the claims made in the main paper.

### E.1  ABLATION STUDIES

In the main paper, we benchmark the Rodrigues Network against several baselines for motion prediction in Cartesian space using a fixed training set of $10^5$ trajectories. Our method demonstrates a significant performance advantage over all baselines. To investigate the contributions of different components of our model, we perform an ablation study by individually removing the Rodrigues Layer, Joint Layer, or Self-attention Layer from the original architecture, and evaluate the resulting performance changes.

The results are presented in Table 11. First, we observe that the Self-attention Layers comprise more than half of the model parameters. However, removing them results in only a slight increase in training error and even a slight decrease in test error. This suggests that while the Self-attention Layer enhances the model's ability to fit the training data, it may slightly hinder generalization. Nonetheless, we retain this component in our default architecture to ensure sufficient model capacity for more complex tasks. Second, the Joint Layers contribute negligibly to the overall parameter count, yet their removal consistently degrades both training and test performance. This highlights their critical role in the effectiveness of the Rodrigues Network. Third, the Rodrigues Layers constitute a substantial portion of the model's parameters. Removing them causes the greatest drop in both training and test performance among all ablation settings. This indicates that the Rodrigues Layer is the most critical component for the model's success. Since the Rodrigues Layer encodes the core inductive bias of our architecture, these findings strongly support our central claim: embedding structural prior of articulated kinematics directly into the network architecture improves learning the actions and motions of articulated actors.

### E.2  TUNING THE BASELINES FOR MOTION PREDICTION

While our method outperforms all baselines on the motion prediction task under the 3-million-parameter setting, we further investigate whether these baselines have been sufficiently tuned. Therefore, we construct three additional configurations for each of the four baselines (GCN, MLP, BoT, and Transformer) with approximately 1M, 10M, and 30M parameters.

The results are shown in Figure 6. For all baselines, the training error consistently decreases as model size increases. However, both GCN and MLP begin to overfit with larger models, as indicated by rising test errors. In contrast, BoT and Transformer exhibit test error saturation, but not overfitting, as parameter count increases to 30M. Importantly, across all parameter scales, our 3M-parameter model outperforms all baseline configurations on both training and test errors. This confirms that the benchmark results reported in the main paper are reliable and that the baselines have been appropriately tuned. Furthermore, even the best-performing baseline (Transformer) with 30M parameters yields a training error that remains higher than the test error of our 3M model. This highlights the superior fitting capability, generalization performance, and parameter efficiency of our proposed approach.

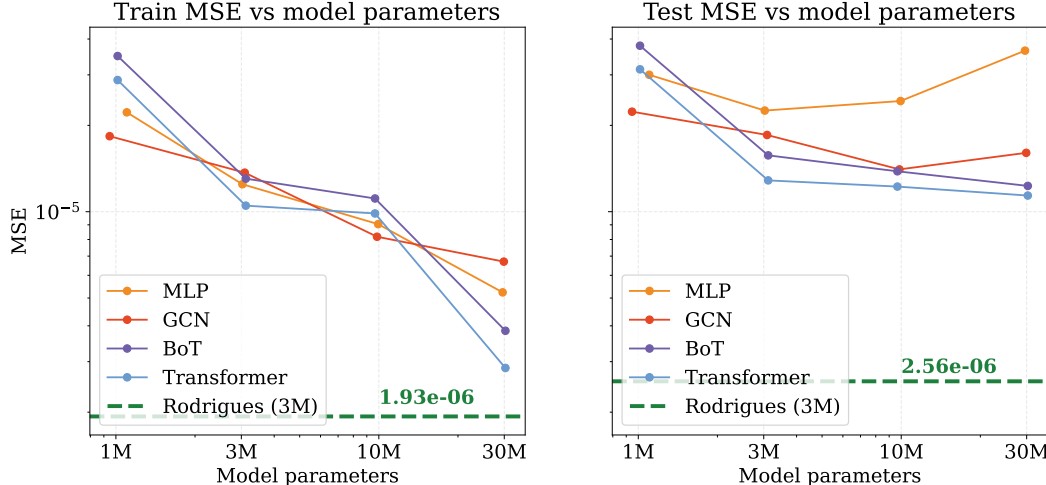

Figure 6: Comparing our method to different baseline configurations in motion prediction with trainset size $= 10^5$.

Table 12: **Hyperparameter sensitivity analysis for the Rodrigues Network on motion prediction in Cartesian space with trainset size** $= 10^5$.

| Variation | $C_J$ | $C_L$ | $B$ | Params (M) | Train MSE (1e-6) | Test MSE (1e-6) |
|---|---|---|---|---|---|---|
| Default | 4 | 8 | 12 | 3.04 | 1.93 | 2.56 |
| Joint channels | 2 | | | 2.43 | 2.18 | 3.00 |
| Joint channels | 8 | | | 4.26 | 1.32 | 1.96 |
| Link channels | | 4 | | 1.19 | 3.64 | 4.35 |
| Link channels | | 16 | | 8.73 | 1.51 | 2.18 |
| Num blocks | | | 6 | 1.55 | 2.24 | 2.76 |
| Num blocks | | | 24 | 6.03 | 3.47 | 4.15 |

### E.3 HYPERPARAMETER SENSITIVITY ANALYSIS FOR THE RODRIGUES NETWORK

We investigate the robustness of our network's performance with respect to variations in key architectural hyperparameters. Specifically, we focus on three core components of the Rodrigues Network: the number of joint channels ($C_J$), link channels ($C_L$), and the number of Rodrigues Blocks ($B$). Starting from the default configuration used in the main paper, we create six variants by either halving or doubling the value of each hyperparameter. We then evaluate these configurations on the motion prediction task to determine whether the network maintains stable performance across these variations. This analysis helps assess the sensitivity of our model to design choices and the general robustness of its architecture.

The results, summarized in Table 12, show that increasing the number of joint channels from 2 to 8 consistently improves both training and test errors. A similar trend is observed for link channels, where increasing from 4 to 16 leads to better performance. In contrast, increasing the number of Rodrigues Blocks from 6 to 24 initially improves train and test errors, but further depth degrades them both, indicating optimization challenges in deeper configurations. Overall, across all hyperparameter variations (up to 4× changes), performance remains within a reasonable range, demonstrating the robustness of our architecture. Furthermore, all variants significantly outperform the strongest baseline from the main paper, suggesting that our method is not only robust but also easily tunable for high performance.

## F  ACCELERATING THE MULTI-CHANNEL RODRIGUES OPERATOR WITH CUDA

The Multi-Channel Rodrigues Operator is a novel component in our architecture, and we found that implementing it directly in PyTorch led to suboptimal speed and memory efficiency. To address this, we developed a custom CUDA kernel that significantly accelerates both the forward and backward passes while reducing memory overhead.

In the forward pass, we assign each CUDA thread block to compute a single output channel over a sub-batch of link features. Each thread accumulates contributions from all input joint and link channels to produce the output feature. This design eliminates the need to explicitly compute and store intermediate tensors such as $U$ and $\bar{U}$ (see Equation 6 in the main paper), thereby saving memory bandwidth and improving runtime performance.

The backward pass adopts a similar strategy of accumulation to avoid intermediate computations. However, it additionally requires explicit gradient computations for both the input joint features and the Rodrigues Kernels, which we handle within the same memory-efficient framework.

This CUDA implementation significantly improves the efficiency of our experiments. For instance, training a 12-block, 16-DoF Rodrigues Network composed solely of Rodrigues Layers, with 16 joint channels and 16 link channels (approximately 52 million parameters), for 100,000 iterations at a batch size of 1024, would take over 100 hours using our PyTorch implementation on a Quadro RTX 6000 GPU. In contrast, using our CUDA kernel to compute the Multi-Channel Rodrigues Operator reduces the training time to approximately 15 hours (over 6x speed-up).

Nonetheless, we acknowledge that existing operators (such as graph convolutions, temporal convolutions, and multi-head attention) have benefited from extensive optimization over the years. Our current implementation does not yet reach that level of maturity. We will release the CUDA source code upon acceptance and hope it serves as a foundation for further research into optimized implementations of the Multi-Channel Rodrigues Operator.

