# OpenReview forum: "Rodrigues Network for Learning Robot Actions"
_ICLR.cc/2026/Conference — ICLR 2026 Oral_

### Official Review · Reviewer_wx1x · 2025-10-26

**Soundness:** 2
**Presentation:** 3
**Contribution:** 2
**Rating:** 2
**Confidence:** 4

**Summary:**

This paper proposes Rodrigues network (rodrinet), a novel graph neural network architecture that encodes the kinematics-aware inductive bias. The authors evaluate the proposed architecture on forward/inverse kinematic fitting, robot manipulation, and hand reconstruction tasks, which show that rodrinet outperforms or is on par with existing popular architectures.

**Strengths:**

1. The overall presentation is very clear and straightforward, give enough background to understand the proposed method.
2. The forward/inverse kinematic fitting experiments clearly demonstrate that the proposed architecture captures the inductive bias of articulated chains. Before that, I hesitated on this because GNN message passing is very different from sequential forward kinematic computation.
3. The Rodriguez equation is an interesting perspective, and the authors conduct experiments over a diverse range of tasks. I very appreciate the authors' efforts.

**Weaknesses:**

My major concerns (questions) are two below, which also link to the core weaknesses of this paper in my opinion.

1. Why robot-kinematic-inductive bias matters?
2. Does the performance gain from RodriNet justify the cost of an unorthodox design?

[Q1]: Most of the current robotic applications use task space control (i.e., predicting the end-effector pose), aka the eef pose, which can be easily translated to joint torques with an op-space controller. In this case, the model does not need to consider the robot's own kinematic structure and can leave that to controller.

In this paper's experiment results, the most compelling results come from the forward/inverse kinematics fitting, which, however, has faster and flexible analytical solutions. On other tasks, the benefits are very marginal. Like in the robot manipulation experiments, even when Rodrinet outperforms the diffusion policy on PickCube and StackCube, I am not entirely convinced because these two tasks should not heavily depend on the robot kinematic chain. In the hand reconstruction task, the authors explicitly mention that modifications are made to RodriNet to suit MANO's configuration space (for example, fitting the hand shape beta parameters, which I assume are not tied to joint/link). This even make the results look less convincing.


[Q2] Even if robot-kinematic-inductive bias may not matter in every task, having them for free does not hurt. However, this proposed architecture, which looks like another GNN variant, has limitations and cannot be used for free.

For example, all robot manipulation experiments are conducted in state space, rather than using visual inputs. This is a critical constraint. Additionally, using a GNN instead of a general transformer could make it challenging to interpolate with other transformer-based backbones. Using joint space and not task space actions can also make the prediction less visually grounded. In sum, these constraints outweigh the marginal performance gain.

A side note: joint controllers are widely used in dexterous hands; maybe this technique can find its place in hand manipulation tasks, instead of parallel grippers.

**Questions:**

(I list all questions in the weakness section)

---

> ### Author Response · Authors · 2025-11-28
> **Response (1/2)**
>
> We sincerely appreciate the reviewer’s thoughtful feedback and the opportunity to clarify our motivation, architectural considerations, and performance benefits. We are encouraged that they find our presentation clear and our experiment range diverse. Below, we address their concerns in detail.
>
> **Q1**: Why robot-kinematic-inductive bias matters?
>
> **[Motivation]**
>
> Task space controllers are limited to robotic arms. But as the robotics community rapidly expands its focus to humanoid, quadruped, and dexterous hands, we believe facilitating action learning in joint-space will be increasingly (if not already) important. Moreover, as discussed in Section 2, articulation-aware inductive biases have been a long-standing interest in CV and character animation (e.g. DiffPose [CVPR 2023], EfficientGCNv1 [T-PAMI 2022] etc), and they are gaining attention in robotics as well (e.g. BoT [CoRL 2024], GET-Zero [ICRA 2025], MI-HGNN [ICRA 2025 etc]). Our work builds on this line of interest.
>
> **[Experimental Evidence]**
>
> First, we would like to clarify that the motion prediction experiment in Section 5.1 does not reduce to simple IK fitting. We designed this task to cleanly compare architectural expressivity and generalization (Line 323).
>
> On the other hand, even for a seemingly trivial computation such as forward kinematics, standard architectures like MLPs and Transformers struggle to fit the mapping. This suggests that these **established architectures are unsuitable for action learning**.
>
> Furthermore, the hand reconstruction experiment is evaluated on a very competitive benchmark against strong and established benchmarks. Improvements on this benchmark are typically small, yet our method achieves gains comparable to those HaMeR achieved over its predecessors, while using only one-fourth of HaMeR’s parameters. We view this not as a marginal improvement, but as **strong evidence of the effectiveness of our network**.
>
> Additionally, (as elaborated in Section A of the supplementary material) the modifications to suit MANO's configuration space **are actually applied to the hand pose theta parameters**. Because each MANO joint is modeled as a full 3-DoF rotation (as opposed to the 1-DoF hinge joints common in robotic hands), we derived a corresponding Neural Rodrigues Operator for full rotations (derivation provided in Section A). This setting **does not “make our results less convincing”**, but rather highlights our flexibility towards both hinge-joint (robotic) and ball-joint (character) kinematics. Moreover, in Table 3, the PA-MPJPE metric depends solely on the theta parameters and is independent of the beta parameters. Our network’s improvement on this metric further supports the effectiveness of our inductive bias. We appreciate the reviewer’s indication that our statement in Line 451 could be misinterpreted, and we will revise it for clarity.
>
> Finally, we agree with the reviewer's side note that dexterous hands are good use cases for our method. Unfortunately, large-scale, reliable imitation-learning benchmarks for dexterous manipulation are still underdeveloped. We therefore selected the ManiSkill suite in Section 5.2 because it is well-established thus enabling meaningful comparisons. Meanwhile, our hand reconstruction experiment already **demonstrates the method’s suitability for dexterous human hands**. Together, we believe these experiments show strong potential for our kinematics-aware approach.

---

> ### Author Response · Authors · 2025-11-28
> **Response (2/2)**
>
> **Q2**: Does the **performance gain** from RodriNet justify the **cost** of an **unorthodox design**?
>
> **[Concerns on cost/limitation]**
>
> First, we argue that our method is **not restricted to state-based inputs**. The hand reconstruction experiment in Section 5.3, where the network takes an RGB image as input and outputs a MANO configuration (Line 431), demonstrates that our architecture effectively supports cross-modal reasoning between visual inputs and joint-space outputs.
>
> Second, we argue that our architecture **does not hinder integration with Transformer-based backbones**. In Section 5.3, we show good interplay with a Vision Transformer encoder by inserting a cross-attention layer into each Rodrigues Block (Line 453). A more detailed explanation is in the supplementary material (Lines 772-782). We will emphasize this capability in the main text based on the reviewer’s concern.
>
> Moreover, a core motivation for adopting a joint-space learning scheme is to avoid the constraints of task-space designs typically tailored for robot arms. Our design and experiments prove that RodriNet is highly versatile, rather than limited by its inductive biases.
>
> **[Performance gains]**
>
> (Discussed in question 1)
>
> **[Unorthodox design]**
>
> Our Neural Rodrigues Operator emerges naturally from the computation structure of articulated kinematics. Both intuition and experimental evidence justify its suitability in learning articulated actions in joint space. It is not only distinctly different from generic message passing as in GNNs, but also well-grounded in kinematic principles.
>
> Moreover, we believe it is beneficial for the robot learning community to explore diverse architectural designs, especially when this new design remains broadly applicable and demonstrates clear advantages. Therefore, we intentionally chose not to restrict ourselves to existing “orthodox” architectures.

---

### Official Review · Reviewer_4VZY · 2025-10-26

**Soundness:** 3
**Presentation:** 3
**Contribution:** 3
**Rating:** 6
**Confidence:** 3

**Summary:**

This paper introduces Rodrigues Network (RodriNet), a neural architecture that explicitly embeds articulated kinematic structure into its design. The key component is the Neural Rodrigues Operator, derived from Rodrigues’ rotation formula by separating “state-dependent” (sin θ, cos θ) terms from “structure-dependent” coefficients and making the latter learnable. This transforms parent–child relations in a kinematic tree into a learnable message-passing rule. A multi-channel variant updates per-link 4×4 features from per-joint inputs using both left and right multiplications. Stacking these operators forms Rodrigues Layers, complemented by a Joint Layer and a Self-Attention Layer for global interaction, with an optional global token handling shared variables such as base pose or gripper state.
Empirically, RodriNet (1) achieves superior fitting accuracy on forward kinematics for the LEAP Hand and improved Cartesian-trajectory prediction on the UR5 arm, (2) boosts simulated success rates in Diffusion Policy across five ManiSkill tasks, and (3) improves single-image 3D hand reconstruction on FreiHAND by replacing the transformer head in HaMeR, yielding small but consistent SOTA gains with significantly fewer parameters.

**Strengths:**

- Principled inductive bias: The formulation of a learnable operator directly from Rodrigues’ rotation rule is mathematically clean and physically grounded. By explicitly modeling the basis (1, sin θ, cos θ), RodriNet captures rotational structure while preserving the kinematic tree topology.

- Strong expressivity for kinematic mapping: On forward-kinematics fitting, RodriNet (Rodrigues Layers only) achieves orders-of-magnitude lower MSE and faster convergence than MLP, GCN, Transformer, or BoT baselines, demonstrating that the inductive bias aligns well with articulated geometry.

- Effective bridging of Cartesian and joint spaces: The UR5 trajectory prediction task is a well-controlled test of inverse kinematics. RodriNet achieves lower train/test MSE and maintains accuracy under reduced data regimes, showing improved data efficiency.

- Practical gains in imitation learning: Integrated into Diffusion Policy, RodriNet improves average success rate (0.61 vs. 0.58 for UNet-DP and 0.44 for Transformer-DP) across five ManiSkill tasks, with the largest gains on PickCube and StackCube. The global token cleanly handles gripper control.

- Cross-domain transfer: When replacing the transformer head in HaMeR, RodriNet attains consistent SOTA on FreiHAND with far fewer parameters (10.7 M vs. 39.5 M), indicating that the inductive bias generalizes beyond robotics.

- Thorough analysis and solid engineering: Ablations identify the Rodrigues Layer as the main contributor. Performance is stable under architectural scaling, and the custom CUDA implementation yields ~6× speedup in large configurations.

**Weaknesses:**

- Lack of equivariance guarantees: The learnable operator applies unconstrained 4×4 left/right multiplications, offering no guarantee of SE(3) consistency or frame equivariance. Layer composition may not preserve physically valid transformations. The method serves as an inductive bias rather than a structured constraint; adding formal regularization (e.g., orthogonality or SE(3)-aware constraints) could strengthen the theoretical foundation.

- Limited novelty relative to existing basis encodings: The operator performs a learned linear combination over {1, sin θ, cos θ} (and quadratic quaternion terms for MANO), similar in spirit to Fourier or harmonic encodings. The absence of comparisons to SE(3)-equivariant architectures (e.g., SE(3)-Transformer, EGNN) or differentiable FK layers leaves it unclear whether gains arise from kinematic-tree structure or from providing the correct trigonometric basis.

- Restricted embodiment generalization: Kernels are defined per joint within a fixed kinematic tree, and experiments are conducted per robot (LEAP Hand, UR5, Franka). There is no evaluation of cross-morphology transfer, shared parameters across repeated structures, or conditioning on structural metadata for multi-embodiment generalization.

- No real-world validation: Results in imitation learning are limited to simulation. For contact-rich tasks such as PegInsertionSide or PlugCharger, performance gains are minimal, suggesting limited robustness to real sensor or actuation noise.

- Compute tradeoffs insufficiently analyzed: Training RodriNet for motion prediction requires longer runtime (2 h 22 m vs. ~1 h 20 m for Transformer/BoT), yet compute–accuracy tradeoffs are not fully discussed.

- Scope limitations: The model handles only rotational joints and omits link geometry or prismatic motion. Its focus on imitation learning rather than closed-loop control narrows applicability to contact-rich or mobile-manipulation domains.

**Questions:**

- Equivariance and structure validity: Does RodriNet satisfy any formal invariance or equivariance guarantees under changes of the base frame or re-rooting of the kinematic tree? Could initializing weights from analytical FK coefficients and applying soft orthogonality constraints improve SE(3) consistency while maintaining flexibility?

- Comparison to SE(3)-aware models: How does RodriNet perform against SE(3)-equivariant architectures (e.g., SE(3)-Transformer, E3NN) or MLP/Transformer baselines with Fourier angle encodings and identical kinematic-graph wiring? This would clarify whether the Neural Rodrigues Operator contributes beyond periodic encodings and structural locality.

- Cross-morphology generalization: Can the per-joint kernels be shared across joints of the same type or conditioned on metadata (axis, link transforms) to enable zero-shot transfer to unseen kinematic trees? A small multi-embodiment study would help test generality.

- Feature interpretation: Do the learned 4×4 “link features” converge toward SE(3)-like structures in practice? Visualizing the 3×3 rotation blocks (e.g., via polar decomposition) could reveal whether the network implicitly maintains orthogonality.

- Compute tradeoffs: RodriNet requires longer training time in the UR5 task. With the CUDA kernel acceleration, what is the effective wall-clock speedup compared to a Transformer at matched accuracy? Are there memory or scaling limits for large CL/CJ?

- Simulation-to-real transfer: In tasks where simulated gains are limited (e.g., PegInsertionSide, PlugCharger), how does RodriNet handle tactile or force-feedback inputs? Any early evidence of transfer performance on hardware would strengthen the empirical case.

---

> ### Author Response · Authors · 2025-12-03
> **Response (1/2)**
>
> We thank the reviewer for the thoughtful review. We are especially encouraged for their recognition of our mathematical formulation and solid engineering. Below, we discuss each question in detail.
>
> **Q1**: Equivariance and structure validity.
>
> We agree that SE(3)-equivariance could be important in certain task setups, but it is not the primary focus of our work. There also widely exist settings where full SE(3) equivariance is not the primary factor that determines performance, and we are mainly focusing on those in this work. For example, as shown in our experiments, the motion prediction and imitation learning tasks with fixed-base robot arms, or the hand reconstruction task with 2D visual inputs.
>
> Although our current Rodrigues Network is not designed to address the problem of SE(3)-equivariance, our key design, the Rodrigues layer, can be easily modified to incorporate equivariance: Observe the double-sided multiplication in the Multi-Channel Rodrigues Operator (Equation 8), it deviates from SE(3)-equivariance only by the left multiplication. So a simple and extremely easy way to incorporate equivariance is to remove the left multiplication and make it a single-sided Rodrigues Layer.
>
> More concrete explanations are as follows: If all input features are left-multiplied by the same 4×4 transformation matrix, all outputs transform accordingly. Formally,
> $$
> \nonumber
> F^{\rm{out}}[j]=\sum _{i=1}^{C_L}F^{\rm {in} }[i]U[i,j] \quad
> \forall 1\le i\le D+1,F^{\rm {in}'}[i]=TF^{\rm {in}}[i] \\\\
> $$
> $$
> \nonumber
> \implies F^{\rm {out}'}[j]=\sum _{i=1}^{C_L}F^{\rm {in}'}[i]U[i,j]=\sum _{i=1}^{C_L}TF^{\rm {in}}[i]U[i,j]=T\sum _{i=1}^{C_L}F^{\rm {in}}[i]U[i,j]=TF^{\rm {out}}[j]
> $$
> In our current design, as equivariance is not the key problem we wanted to tackle, we use double-sided multiplication for better expressivity. To show the difference, here we provide an additional ablation comparison for the double-sided vs single-sided on the motion prediction task.
>
> | **Backbones**            | **error_ee_trans (mm)** | **error_ee_rot (˚)** | **error_theta (˚)** | **MSE (\*1e-6)** | **train MSE (\*1e-6)** |
> | ------------------------ | ----------------------- | -------------------- | ------------------- | ---------------- | ---------------------- |
> | RodriguesMotion          | 1.21                    | 0.16                 | 0.06                | 2.56             | 1.93                   |
> | RodriguesMotion (single) | 1.35                    | 0.16                 | 0.07                | 3.01             | 2.35                   |
>
> We would like to thank the reviewer for raising this point. Although equivariance is not the main theme of our paper, we believe that the easy switches between non-equivariance and equivariance in our network layer open a door for interesting future directions, such as a unified framework that can adapt to different levels of equivariance according to different settings.
>
>
>
> **Q2**: Comparison to SE(3)-aware models
>
> We thank the reviewer again for the suggested comparisons. Here we constructed new BoT and Transformer baselines enhanced with Fourier angle encodings (16 frequency levels), and benchmarked them on the motion prediction experiment. The results are as follows:
>
> | **Backbones**    | **error_ee_trans (mm)** | **error_ee_rot (˚)** | **error_theta (˚)** | **MSE (\*1e-6)** | **train MSE (\*1e-6)** |
> | ---------------- | ----------------------- | -------------------- | ------------------- | ---------------- | ---------------------- |
> | RodriguesMotion  | 1.21                    | 0.16                 | 0.06                | 2.56             | 1.93                   |
> | FourierTrMotion  | 4.41                    | 0.58                 | 0.20                | 36.37            | 11.89                  |
> | FourierBoTMotion | 4.52                    | 0.59                 | 0.21                | 34.74            | 16.88                  |
>
> In summary, these Fourier angle encoding baselines perform significantly worse than our network. This indicates that periodicity handling and SE(3)-awareness alone are insufficient. To get better performance in robot action learning, kinematics-aware inductive bias is needed.
>
> Regarding SE(3)-Transformers: after careful consideration, we believe their relative-position technique assumes access to the 3D spatial coordinates of nodes/links. This assumption is correct in their paper’s applications, like n-body simulation and molecular property prediction. On the other hand, robot action learning does not typically assume known 3D link coordinates, and the network must instead infer the articulated configuration. For example, in the hand reconstruction experiment, the model only takes a 2D image and must predict the 3D hand pose. Because of this mismatch in assumed inputs, we do not find a straightforward or meaningful way to compare against SE(3)-Transformers. This also highlights that our formulation is more versatile.

---

> ### Author Response · Authors · 2025-12-03
> **Response (2/2)**
>
> **Q3**: Cross-morphology generalization.
>
> One motivation of our design is to bring more embodiment information into the learning framework. Our current design of the Rodrigues Kernels is joint-specific (but exact 3D spatial coordinates of links and joints are not required). This choice is for two reasons:(1) theoretical grounding: each Rodrigues Rotation Formula depends on a particular joint’s state-independent parameters, (2) in practical settings, the robot’s embodiment is typically known and fixed.
>
> We agree with the reviewer that a unified framework for different embodiments would be good. For example, a conditioned Rodrigues Network whose kernels can flexibly condition on additional embodiment inputs. However, at the current moment, cross-embodiment is a highly non-trivial problem even for the entire field. The difficulty is not just about network designs, but also involves many other aspects, such as data, problem formulations, training objectives, etc. For example, to achieve zero-shot cross-morphology generalisation, one would require a large and diverse database of joint morphologies, which is not well-established for robot learning at the current moment.
>
> We agree that a cross-morphology Neural Rodrigues Operator would certainly be an interesting direction. In fact, we believe that for cross-embodiment, when other difficulties are better resolved in the future (e.g., data, training objectives, etc.), how to efficiently incorporate embodiment information into the network may become one of the most important problems – where we believe that our Rodrigues network would have additional potential.
>
> **Q4**: Feature interpretation:
>
> We do not find that the 4x4 link features converge towards SE(3)-like structures in practice, which aligns with our intentions that the Neural Rodrigues Operator should "encode richer, higher level features beyond joint angles and link poses" (Line 181). This makes features more flexible and expressive, though harder to interpret.
>
> To provide other possible insight, we visualized the attention maps of the Self-Attention Layer in the motion prediction experiment, and observed that the value corresponding to the end-effector link’s feature token as the attention target is often significantly higher than other values. This indicates that the attention head is “moving” information from a link toward the end-effector, potentially capturing a task-specific information flow along the kinematic chain.
>
> **Q5**: Compute tradeoffs?
>
> As shown in Figure 3 (b), the Rodrigues Network converges significantly faster in terms of iteration count in the fk experiment.
>
> Regarding wall-clock time, in the motion prediction experiment, the Transformer baseline reaches its best test loss of 12.86e-6 after 100k iterations, taking 80 minutes. To reach a comparable test loss, our Rodrigues Network requires less than 30k iterations, taking 43 minutes, reaching 11.94e-6 test loss, corresponding to an approximate wall-clock speedup of 2x.
>
> In terms of compute, memory usage grows linearly with respect to the number of DoFs and linearly with respect to CJ, but quadratically with respect to CL.
>
> We also note that our current CUDA implementation still has room for optimization, and expect further speedups with kernel-level tuning. On the other hand, Transformer variants and other established neural operators benefit from substantial years-long engineering optimizations.
>
> **Q6**: Simulation-to-real transfer:
>
> The ManiSkill benchmark does not provide tactile or force-feedback inputs, so our current experiments do not evaluate any network under these modalities. Our statement in Line 425 was intended to mean that the low success rates observed for all methods might be due to the lack of force inputs, so that the limitation of the architecture is no longer the most critical factor here. We will revise this sentence to reduce ambiguity based on the reviewer’s feedback.

---

### Official Review · Reviewer_LiRT · 2025-10-30

**Soundness:** 3
**Presentation:** 3
**Contribution:** 3
**Rating:** 8
**Confidence:** 4

**Summary:**

The authors investigate incorporating structural inductive biases from kinematic models in neural network design. They propose the Rodriguez layer, which combines projecting joint features through the learned Rodriguez operator and aggregating features globally with self-attention. Experimentally, the authors show that for problems where agents have a rigid body structure (e.g., forward kinematics or solving manipulation tasks with imitation learning), their model design performs substantially better than neural architectures that lack these inductive biases.

**Strengths:**

> Incorporating learnable kinematic models in robotics makes intuitive sense and seems to provide measurable improvements for applying machine learning on rigid body structures

> The paper is well written and presents the proposed model clearly. The diagrams help summarize the approach.

> The experiments show compelling results to validate the efficacy of the Rodriguez layer for rigid body agents. The inclusion of both robots and animated characters helps demonstrate the model's potential across several settings of embodied agents.

**Weaknesses:**

> Some aspects of the work are a bit obvious. The results for forward kinematics are not surprising, particularly as the Rodriguez matrix is one of two major approaches to modelling rigid-body kinematics—the other being Denavit-Hartenberg parameters. It could strengthen the paper to show how directly predicting the kinematic model's parameters compares in this problem.

> Some results could benefit from statistical hypothesis testing to confirm whether the observed benefits are statistically significant (e.g. the success rates in Table 2).

> Something that might be helpful would be to include results for failure cases. Presumably, there are data modalities or scenarios in which assuming a kinematic relation negatively impacts performance.

Writing Opinions:

Line 063 - 068:  This paragraph can be cut with almost no loss of meaningful content to the paper.

Related Work: Consider looking at cross-embodiment works as potential related work. These models also consider agent structure as necessary to model explicitly.

[1] Gupta, Agrim, et al. "Metamorph: Learning universal controllers with transformers." arXiv preprint arXiv:2203.11931 (2022).

[2] Xiong, Zheng, Jacob Beck, and Shimon Whiteson. "Universal morphology control via contextual modulation." International Conference on Machine Learning. PMLR, 2023.

> Figure 3: We suggest replacing "backbone" with "architecture" unless these models are pretrained, which, in our experience, is where this term is used more frequently.

>line 353: replace backbone with "architecture"

> Figure 5: Provide more information in the caption on what is shown in the diagram.

**Questions:**

Q1: What's the reason for not comparing it to the Denavit–Hartenberg formulation of kinematic structure? A quick search reveals that the Rodrigues formulation does have some technical advantages, but this seems the more appropriate comparison.

Q2: What kind of features are provided to the Rodriguez layers? Do the transformations process non-kinematic related features in some way? How would these layers work with non-kinematic associated features, for instance?

Q3: What about the author's current work that limits their representation to only spherical joints as opposed to translational joints?

Q4: What distinguishes the Rodriguez layer from running system identification to find the model's kinematics?

---

> ### Author Response · Authors · 2025-11-28
>
> We thank the reviewer for the thoughtful feedback. We are encouraged that they find our writing clear and our results compelling. We address the questions in detail below.
>
> **Q1**: What's the reason for not comparing it to the Denavit–Hartenberg formulation of kinematic structure? A quick search reveals that the Rodrigues formulation does have some technical advantages, but this seems the more appropriate comparison.
>
> We agree that DH formulation is a widely used convention for representing and analytically computing forward kinematics. However, it imposes two structural constraints on the coordinate frames: (1) each joint axis must align with the z-axis, (2) the x-axis must be the common normal between consecutive joints. These constraints, while convenient for analytical derivations, make it unclear how to construct a learnable operator that respects them during training. On the other hand, the Rodrigues formulation naturally supports arbitrary joint axes and provides a symmetric and flexible mathematical structure when extended into a learnable neural operator. For these reasons, we focus on the Rodrigues-based architecture rather than DH-based variants.
>
> **Q2**: What kind of features are provided to the Rodriguez layers? Do the transformations process non-kinematic related features in some way? How would these layers work with non-kinematic associated features, for instance?
>
> The core motivation behind the Neural Rodrigues Operator is the assumption that features relevant to robot action learning (whether explicitly kinematic or not) benefit from being transformed in a manner consistent with articulated kinematics. Therefore, the Rodrigues Layers apply the transformation in Equation 8 to all link-associated features. Even when the features are not strictly kinematic, the Rodrigues Layer introduces an articulation-aware inductive bias that improves expressivity and generalization. This hypothesis is supported by the ablation study in Section E.1.
>
> **Q3**: What about the author's current work that limits their representation to only spherical joints as opposed to translational joints?
>
> At present, our Neural Rodrigues Operator is designed to support rotational joints. Additionally, we demonstrate that the Global Token effectively supports a parallel gripper attached to the end of a robotic arm. This design choice reflects the fact that the vast majority of robot embodiments used in contemporary robot learning (robot arms, dexterous hands, humanoids, and quadrupeds) are composed almost entirely of rotational joints (possibly with grippers). Thus, we believe that the absence of translational joint does not significantly limit our method's applicability to robot learning. Focusing on rotational joints also keeps the mathematical formulation and the resulting neural architecture clean and principled. We think extending our operator to fully handle translational joints is a promising direction for future work.
>
> **Q4**: What distinguishes the Rodriguez layer from running system identification to find the model's kinematics?
>
> As discussed in Lines 177-183, the Rodrigues Layer is designed as a general neural operator that embeds a kinematic-driven inductive bias. Its purpose is not to reconstruct the robot’s exact kinematic model, but to provide a versatile neural operator that can support a wide range of learned functions depending on training data. For instance, Section 5.2 demonstrated that it can be used to learn a control policy through imitation learning, which is beyond the capability of classical system identification.
>
> **Writing suggestions**: We will enrich the related work section with the suggested literature and revise the figure captions to reduce ambiguity and provide more informative descriptions.

---

### Official Review · Reviewer_fHxR · 2025-11-01

**Soundness:** 3
**Presentation:** 3
**Contribution:** 3
**Rating:** 8
**Confidence:** 3

**Summary:**

This paper introduces RodriNet, a neural architecture that embeds articulated-kinematics structure directly into its computation by learning a generalized version of Rodrigues’ rotation formula. Instead of treating joint values as flat vectors like MLPs or Transformers, the proposed Neural Rodrigues Operator replaces fixed trigonometric coefficients in classical forward-kinematics equations with learnable weights, enabling structured message passing along a robot's kinematic tree. RodriNet stacks these operators with joint-update and attention layers, yielding a model that naturally learns hierarchical motion patterns. Experiments show large gains in fitting forward kinematics, predicting 3D joint trajectories from Cartesian motions, improving imitation-learning performance when used as the backbone in Diffusion Policy, and achieving state-of-the-art results on 3D hand pose estimation, all with fewer parameters. Overall, the work argues for architectural priors tailored to robot embodiment as a path to more efficient and generalizable action learning.

**Strengths:**

- Introduces a learnable generalization of Rodrigues rotation to embed articulated-kinematics structure into neural networks, giving a principled inductive bias for articulated action learning

- Presents a clear architecture (Rodrigues Layer, Joint Layer, self-attention) that mixes local kinematic structure with global information exchange

- Demonstrates strong improvements across domains: forward kinematics, Cartesian-to-joint motion prediction, robotic imitation learning, and 3D hand reconstruction

- Shows faster convergence and better data efficiency than standard architectures like MLPs, GCNs, and Transformers in modeling articulated motion

**Weaknesses:**

- Requires per-joint learnable parameters and tree-structured computation, which may introduce higher architectural complexity compared to simpler universal backbones

- No ablation studying how much each component (Rodrigues layer vs joint layer vs attention) contributes, leaving uncertainty about which parts drive performance

**Questions:**

-

---

> ### Author Response · Authors · 2025-11-28
>
> We thank the reviewer for the thoughtful and constructive review. We are greatly encouraged by their recognition of our architectural formulation and experimental results. Below, we provide detailed responses to your concerns.
>
> **Concern**: No ablation studying how much each component (Rodrigues layer vs joint layer vs attention) contributes, leaving uncertainty about which parts drive performance.
>
> **Response**: Section E.1 of the supplementary material presents an ablation study in which we remove the Rodrigues Layer, Joint Layer, and Self-Attention Layer individually. The key findings are: 1) the Self-attention Layers improve the model's ability to fit the training data, but might slightly hinder generalization, 2) the Joint Layers, despite contributing negligibly to the parameter count, play an essential role in both training and testing performance; and 3) removing the Rodrigues Layers leads to the largest drop in both training and testing accuracy, indicating that they are the most critical components of the network. We will integrate these results into the main paper to highlight component contributions more clearly based on the reviewer's feedback.
>
> **Concern**: Requires per-joint learnable parameters and tree-structured computation, which may introduce higher architectural complexity compared to simpler universal backbones
>
> **Response**: We agree that architectural complexity is an important practical consideration. In fact, we have reported the training time and hardware specifications for all methods in Tables 9 and 10 of the supplementary material. Additionally, each Rodrigues Layer is designed so that every output link feature retrieves input features only from itself, its parent link, and its parent joint. This design avoids propagating information sequentially along the entire kinematic tree within a single layer. Instead, each layer passes information one step down the tree in parallel, enabling substantial parallelization across links and channels. Therefore, we believe the computation complexity is not significantly high. Furthermore, as described in Section F of the supplementary, we have implemented a CUDA-based version of the Rodrigues Operator to improve efficiency. We believe further optimization remains possible, and we will continue exploring acceleration strategies.

---

### Author Response · Authors · 2025-12-03
**General response**

We thank all the reviewers for their feedback. We also want to particularly thank the AC for their additional time and effort. We will incorporate the feedback into the revision.

First of all, the majority has consensus on the following strengths of our paper:

- Idea: (i) interesting and intuitive, (ii) theoretically grounded.
- Method is useful: (i) good performances, (ii) applicable to diverse domains.
- Experiments are well-designed: thorough, well-controlled, diverse settings.
- The paper is well-written.

Besides, we have provided explanations and additional experimental results to address the reviewers’ questions and concerns. Here is a brief summary: (for more detailed responses to each specific question, please check our response to each individual reviewer)

- Generalization of the network layer design:
  - To more diverse embodiments: (i) other types of joints (LiRT). (ii) unseen morphology (cross-embodiment) (4VZY).
  - Equivariance (4VZY).

We explained how our network has potential for more generalized settings with easy extensions/modifications, and how the current design fits into the needs and barriers of the field at the current moment.

- Computational complexity: we (i) explained our computational scalability (e.g., parallelization) (fHxR); (ii) evaluated our compute-accuracy trade-off (4VZY).
- Experiments:
  - Certain experiments asked by the reviewers already exist in the paper or the supplementary material, and we have pointed the reviewers to the specific sections.
  - Additional comparison: For some new perspectives raised by the reviewers (e.g., equivariance, time vs accuracy), we conducted additional experimental comparisons and analysis.
- Motivation: Reviewer wx1x is unclear about the motivation of the work, we explained it from two perspectives: (i) intuitively, what motivates the joint-space learning designs; (ii) experimentally, what the performance gains are.

---

### Meta-Review · Area_Chair_dkwv · 2026-01-06

**Summary:**

Three reviewers recommend acceptance while one reviewer (wx1x) rejection. The main concerns of wx1x are two-folds: Why robot-kinematic-inductive bias matters? Does the performance gain from RodriNet justify the cost of an unorthodox design? The author rebuttals explain that action learning in joint space is increasingly important and articulation-aware inductive biases have been a long-standing interest in CV and character animation. Also, the proposed method is not limited to state-based inputs. They performed the hand reconstruction experiment taking an RGB image estimating a MANO configuration. Rather than that, the reviewers agree that the work introduces a learnable generalization of Rodrigues rotation to embed articulated-kinematics structure into neural networks, presents a clear NN architecture, and demonstrates strong improvements and faster convergence and data efficiency.

**Reviewer Concerns:**

The same as above.

**Reviewer Scores:**

The four reviewers initially gave 2,8,8,6 respectively. The author rebuttals satisfactorily addressed the main issues of wx1x and the other reviewers.

---

### Decision · Program_Chairs · 2026-01-26

Accept (Oral)